# TabNAS: Rejection Sampling for
# Neural Architecture Search on Tabular Datasets

**Chengrun Yang**[1], **Gabriel Bender**[1], **Hanxiao Liu**[1], **Pieter-Jan Kindermans**[1],
**Madeleine Udell**[2], **Yifeng Lu**[1], **Quoc V. Le**[1], **Da Huang**[1]
{chengrun, gbender, hanxiaol, pikinder}@google.com,
udell@stanford.edu, {yifenglu, qvl, dahua}@google.com

[1] Google Research, Brain Team [2] Stanford University

## Abstract

The best neural architecture for a given machine learning problem depends on many factors: not only the complexity and structure of the dataset, but also on resource constraints including latency, compute, energy consumption, etc. Neural architecture search (NAS) for tabular datasets is an important but under-explored problem. Previous NAS algorithms designed for image search spaces incorporate resource constraints directly into the reinforcement learning (RL) rewards. However, for NAS on tabular datasets, this protocol often discovers suboptimal architectures. This paper develops TabNAS, a new and more effective approach to handle resource constraints in tabular NAS using an RL controller motivated by the idea of rejection sampling. TabNAS immediately discards any architecture that violates the resource constraints without training or learning from that architecture. TabNAS uses a Monte-Carlo-based correction to the RL policy gradient update to account for this extra filtering step. Results on several tabular datasets demonstrate the superiority of TabNAS over previous reward-shaping methods: it finds better models that obey the constraints.

## 1 Introduction

To make a machine learning model better, one can scale it up. But larger networks are more expensive as measured by inference time, memory, energy, etc, and these costs limit the application of large models: training is slow and expensive, and inference is often too slow to satisfy user requirements.

Many applications of machine learning in industry use tabular data, e.g., in finance, advertising and medicine. It was only recently that deep learning has achieved parity with classical tree-based models in these domains [9, 11]. For vision, optimizing models for practical deployment often relies on Neural Architecture Search (NAS). Most NAS literature targets convolutional networks on vision benchmarks [14, 5, 10, 19]. Despite the practical importance of tabular data, however, NAS research on this topic is quite limited [8, 7]. (See Appendix A for a more comprehensive literature review.)

Weight-sharing reduces the cost of NAS by training a *SuperNet* that is the superset of all candidate architectures [2]. This trained SuperNet is then used to estimate the quality of each candidate architecture or *child network* by allowing activations in only a subset of the components of the SuperNet and evaluating the model. Reinforcement learning (RL) has shown to efficiently find the most promising child networks [16, 5, 3] for vision problems.

In our experiments, we show that a direct application of approaches designed for vision to tabular data often fails. For example, the TuNAS [3] approach from vision struggles to find the optimal architectures for tabular datasets (see experiments). The failure is caused by the interaction of the search space and the factorized RL controller. To understand why, consider the following toy example

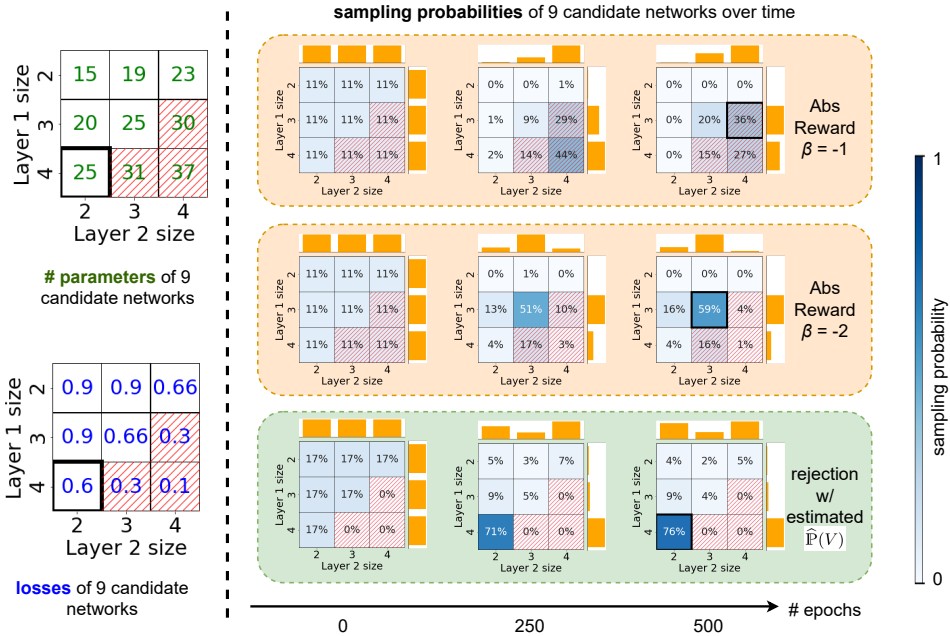

Figure 1: **A toy example for tabular NAS** in a 2-layer search space with a 2-dimensional input and a limit of 25 parameters. Each cell represents an architecture. The left half shows the number of parameters and loss of each candidate in the search space. Infeasible architectures have red-striped cells. The bottom left cell (bold border) is the global optimum with size 4 for the first hidden layer and size 2 for the second. The right half shows the change of sampling probabilities in NAS with different RL rewards. Sampling probabilities are shown both as percentages in cells, and intensity indicated by the right colorbar. Orange bars on the top and right sides show the (independent) sampling probability distributions of size candidates for individual layers. With the Abs Reward, the sampling probability of each architecture is the product of sampling probabilities of each layer; with the rejection-based reward, the probability of an infeasible architecture is 0, and probabilities of feasible architectures are normalized to sum to 1. At epoch 500, the cell squared in bold shows the architecture picked by the corresponding RL controller. RL with the Abs Reward $Q(y) + \beta|T(y)/T_0 - 1|$ proposed in TuNAS [3] either converges to a feasible but suboptimal architecture ($\beta = -2$, middle row) or violates the resource constraint ($\beta = -1$, top row). Other latency-aware reward functions show similar failures. In contrast, TabNAS converges to the optimum (bottom row).

with 2 layers, illustrated in Figure 1. For each layer, we can choose a layer size of 2, 3, or 4, and the maximum number of parameters is set to 25. The optimal solution is to set the size of the first hidden layer to 4 and the second to 2. Finding this solution with RL is difficult with a cost penalty approach. The RL controller is initialized with uniform probabilities. As a result, it is quite likely that the RL controller will initially be penalized heavily when choosing option 4 for the first layer, since two thirds of the choices for the second layer will result in a model that is too expensive. As a result, option 4 for the first layer is quickly discarded by the RL controller and we get stuck in a local optimum.

This co-adaptation problem is caused by the fact that existing NAS methods for computer vision often use factorized RL controllers, which force all choices to be be made independently. While factorized controllers can be optimized easily and are parameter-efficient, they cannot capture all of the nuances in the loss landscape. A solution to this could be to use a more complex model such as an LSTM (e.g., [16, 4]). However, LSTMs are often much slower to train and are far more difficult to tune.

Our proposed method, TabNAS, uses a solution inspired by rejection sampling. It updates the RL controller only when the sampled model satisfies the cost constraint. The RL controller is then discouraged from sampling poor models within the cost constraint and encouraged to sample the high quality models. Rather than penalizing models that violate the constraints, the controller silently discards them. This trick allows the RL controller to see the true constrained loss landscape, in which

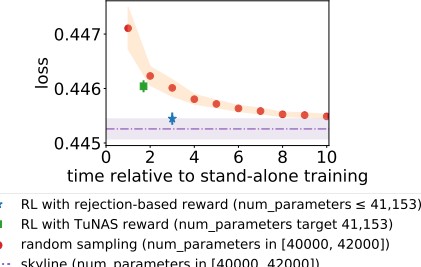

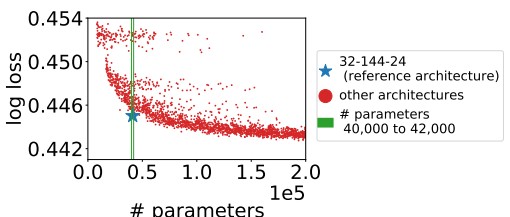

Figure 2: TabNAS reward distributionally outperforms random search and resource-aware Abs Reward on the Criteo dataset within a 3-layer search space. All error bars and shaded regions are 95% confidence intervals. The x axis is the time relative to train time for a single architecture. The y axis is the validation loss. More details in Appendix C.2.3.

Figure 3: Validation loss (logistic) vs. number of parameters on Criteo with a 3-layer search space. The standard deviation (std) of architecture performance for different runs is 0.0002, so architectures whose performance difference is larger than 2std are qualitatively different with high probability. The search space and Pareto-optimal architectures are shown in Appendix C.2.1.

having some large layers is beneficial, allowing TabNAS to efficiently find global (not just local) optima for tabular NAS problems. Our contributions can be summarized as follows:

- We identify failure cases of existing resource-aware NAS methods on tabular data and provide evidence this failure is due to the cost penalty in the reward together with the factorized space.
- We propose and evaluate an alternative: a rejection sampling mechanism that ensures the RL controller only selects architectures that satisfy resource constraint. This extra rejection step allows the RL controller to explore parts of the search space that would otherwise be overlooked.
- The rejection mechanism also introduces a systematic bias into the RL gradient updates, which can skew the results. To compensate for this bias, we introduce a theoretically motivated and empirically effective correction into the gradient updates. This correction can be computed exactly for small search spaces and efficiently approximated by Monte-Carlo sampling otherwise.
- We show the resulting method, TabNAS, automatically learns whether a bottleneck structure is needed in an optimal architecture, and if needed, where to place the bottleneck in the network.

These contributions form TabNAS, our RL-based weight-sharing NAS with rejection-based reward. TabNAS robustly and efficiently finds a feasible architecture with optimal performance within the resource constraint. Figure 2 shows an example.

## 2 Notation and terminology

**Math basics.** We define $[n] = \{1, \cdots, n\}$ for a positive integer $n$. With a Boolean variable $\mathcal{X}$, the indicator function $\mathbb{1}(\mathcal{X})$ equals 1 if $\mathcal{X}$ is true, and 0 otherwise. $|S|$ denotes the cardinality of a set $S$; stop_grad($f$) denotes the constant value (with gradient 0) corresponding to a differentiable quantity $f$, and is equivalent to `tensorflow.stop_gradient(f)` in TensorFlow [1] or `f.detach()` in PyTorch [15]. $\subseteq$ and $\subset$ denote subset and strict subset, respectively. $\nabla$ denotes the gradient with respect to the variable in the context.

**Weight, architecture, and hyperparameter.** We use *weights* to refer to the parameters of the neural network. The *architecture* of a neural network is the structure of how nodes are connected; examples of architectural choices are hidden layer sizes and activation types. *Hyperparameters* are the non-architectural parameters that control the training process of either stand-alone training or RL, including learning rate, optimizer type, optimizer parameters, etc.

**Neural architecture.** A neural network with specified architecture and hyperparameters is called a *model*. We only consider fully-connected feedforward networks (FFNs) in this paper, since they can already achieve SOTA performance on tabular datasets [11]. The number of hidden nodes after each weight matrix and activation function is called a *hidden layer size*. We denote a single network in our search space with hyphen-connected choices. For example, when searching for hidden layer sizes, in the space of 3-hidden-layer ReLU networks, 32-144-24 denotes the candidate where the sizes of

the first, second and third hidden layers are 32, 144 and 24, respectively. We only search for ReLU networks; for brevity, we will not mention the activation function type in the sequel.

**Loss-resource tradeoff and reference architectures.** In the hidden layer size search space, the validation loss in general decreases with the increase of the number of parameters, giving the loss-resource tradeoff (e.g., Figure 3). Here loss and number of parameters serve as two *costs* for NAS. Thus there are Pareto-optimal models that achieve the smallest loss among all models with a given bound on the number of parameters. With an architecture that outperforms others with a similar or fewer number of parameters, we do resource-constrained NAS with the number of parameters of this architecture as the resource target or constraint. We call this architecture the *reference architecture* (or *reference*) of NAS, and its performance the *reference performance*. We do NAS with the goal of *matching* (the size and performance of) the reference. Note that the RL controller only has knowledge of the number of parameters of the reference, and is not informed of its hidden layer sizes.

**Search space.** When searching $L$-layer networks, we use capital letters like $X = X_1\text{-}\cdots\text{-}X_L$ to denote the random variable of sampled architectures, in which $X_i$ is the random variable for the size of the $i$-th layer. We use lowercase letters like $x = x_1\text{-}\cdots\text{-}x_L$ to denote an architecture sampled from the distribution over $X$, in which $x_i$ is an instance of the $i$-th layer size. When there are multiple samples drawn, we use a bracketed superscript to denote the index over samples: $x^{(k)}$ denotes the $k$-th sample. The search space $S = \{s_{ij}\}_{i\in[L],j\in[C_i]}$ has $C_i$ choices for the $i$-th hidden layer, in which $s_{ij}$ is the $j$-th choice for the size of the $i$-th hidden layer: for example, when searching for a one-hidden-layer network with size candidates {5, 10, 15}, we have $s_{13} = 15$.

**Reinforcement learning.** The RL algorithm learns the set of logits $\{\ell_{ij}\}_{i\in[L],j\in[C_i]}$, in which $\ell_{ij}$ is the logit associated with the $j$-th choice for the $i$-th hidden layer. With a fully factorized distribution of layer sizes (we learn a separate distribution for each layer), the probability of sampling the $j$-th choice for the $i$-th layer $p_{ij}$ is given by the SoftMax function: $p_{ij} = \exp(\ell_{ij})/\sum_{j\in[C_i]}\exp(\ell_{ij})$. In each RL step, we sample an architecture $y$ to compute the single-step RL objective $J(y)$, and update the logits with $\nabla J(y)$: an unbiased estimate of the gradient of the RL value function.

**Resource metric and number of parameters.** We use the number of parameters, which can be easily computed for neural networks, as a cost metric in this paper. However, our approach does not depend on the specific cost used, and can be easily adapted to other cost metrics.

## 3 Methodology

Our NAS methodology can be decomposed into three main components: weight-sharing with layer warmup, REINFORCE with one-shot search, and Monte Carlo (MC) sampling with rejection.

As an overview, our method starts with a SuperNet, which is a network that layer-wise has width equal to the largest choice within the search space. We first stochastically update the weights of the entire SuperNet to "warm up" over the first 25% of search epochs. Then we alternate between updating the shared model weights (which are used to estimate the quality of different child models) and the RL controller (which focuses the search on the most promising parts of the space). In each iteration, we first sample a child network from the current layer-wise probability distributions and update the corresponding weights within the SuperNet (weight update). We then sample another child network to update the layerwise logits that give the probability distributions (RL update). The latter RL update is only performed if the sampled network is feasible, in which case we use rejection with MC sampling to update the logits with a sampling probability conditional on the feasible set.

To avoid overfitting, we split the labelled portion of a dataset into training and validation splits. Weight updates are carried out on the training split; RL updates are performed on the validation split.

### 3.1 Weight sharing with layer warmup

The weight-sharing approach has shown success on various computer vision tasks and NAS benchmarks [16, 2, 5, 3]. To search for an FFN on tabular datasets, we build a SuperNet where the size of each hidden layer is the largest value in the search space. Figure 4 shows an example. When we sample a child network with a hidden layer size $\ell_i$ smaller than the SuperNet, we only use the first $\ell_i$ hidden nodes in that layer to compute the output in the forward pass and the gradients in the

backward pass. Similarly, in RL updates, only the weights of the child network are used to estimate the quality reward that is used to update logits.

In weight-sharing NAS, warmup helps to ensure that the SuperNet weights are sufficiently trained to properly guide the RL updates [3]. With probability $p$, we train all weights of the SuperNet, and with probability $1 - p$ we only train the weights of a random child model. When we run architecture searches for FFNs, we do warmup in the first 25% epochs, during which the probability $p$ linearly decays from 1 to 0 (Figure 5(a)). The RL controller is disabled during this period.

## 3.2    One-shot training and REINFORCE

We do NAS on FFNs with a REINFORCE-based algorithm. Previous works have used this type of algorithm to search for convolutional networks on vision tasks [18, 5, 3]. When searching for $L$-layer FFNs, we learn a separate probability distribution over $C_i$ size candidates for each layer. The distribution is given by $C_i$ logits via the SoftMax function. Each layer has its own independent set of logits. With $C_i$ choices for the $i$th layer, where $i = 1, 2, \cdots, L$, there are $\prod_{i \in [L]} C_i$ candidate networks in the search space but only $\sum_{i \in [L]} C_i$ logits to learn. This technique significantly reduces the difficulty of RL and make the NAS problem practically tractable [5, 3].

The REINFORCE-based algorithm trains the SuperNet weights and learns the logits $\{\ell_{ij}\}_{i \in [L], j \in [C_i]}$ that give the sampling probabilities $\{\ell_{ij}\}_{i \in [L], j \in [C_i]}$ over size candidates by alternating between weight and RL updates. In each iteration, we first sample a child network $x$ from the SuperNet and compute its training loss in the forward pass. Then we update the weights in $x$ with gradients of the training loss computed in the backward pass. This weight update step trains the weights of $x$. The weights in architectures with larger sampling probabilities are sampled and thus trained more often. We then update the logits for the RL controller by sampling a child network $y$ that is independent of the network $x$ from the same layerwise distributions, computing the quality reward $Q(y)$ as $1 - loss(y)$ on the validation set, and then updating the logits with the gradient of $J(y) = \text{stop\_grad}(Q(y) - \bar{Q}) \log \mathbb{P}(y)$: the product of the advantage of $y$'s reward over past rewards (usually an exponential moving average) and the log-probability of the current sample.

The alternation creates a positive feedback loop that trains the weights and updates the logits of the large-probability child networks; thus the layer-wise sampling probabilities gradually converge to more deterministic distributions, under which one or several architectures are finally selected.

Details of a resource-oblivious version is shown as Appendix B Algorithm 1, which does not take into account a resource constraint. In Section 3.3, we show an algorithm that combines Monte-Carlo sampling with rejection sampling, which serves as a subroutine of Algorithm 1 by replacing the probability in $J(y)$ with a conditional version.

## 3.3    Rejection-based reward with MC sampling

Only a subset of the architectures in the search space $S$ will satisfy resource constraints; $V$ denotes this set of feasible architectures. To find a feasible architecture, a resource target $T_0$ is often used in an RL reward. Given an architecture $y$, a resource-aware reward combines its quality $Q(y)$ and resource consumption $T(y)$ into a single reward. MnasNet [18] proposes the rewards $Q(y)(T(y)/T_0)^\beta$ and $Q(y) \max\{1, (T(y)/T_0)^\beta\}$ while TuNAS [3] proposes the absolute value reward (or Abs Reward) $Q(y) + \beta|T(y)/T_0 - 1|$. The idea behind is to encourage models with high quality with respect the resource target. In these rewards $\beta$ is a hyperparameter that needs careful tuning.

We find that on tabular data, RL controllers using these resource-aware rewards above can struggle to discover high quality structures. Figure 1 shows a toy example in the search space in Figure 4, in which we know the validation losses of each child network and only train the RL controller for 500 steps. The optimal network is 4-2 among architectures with number of parameters no more than 25, but the RL controller rarely chooses it. In Section 4.1, we show examples on real datasets.

This phenomenon reveals a gap between the true distribution we want to sample from and the distributions obtained by sampling from this factorized search space:

- We *only* want to sample from the set of feasible architectures $V$, whose distribution is $\{\mathbb{P}(y \mid y \in V)\}_{y \in V}$. The resources (e.g., number of parameters) used by an architecture, and thus its feasibility, is determined jointly by the sizes of all layers.

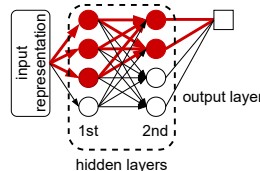

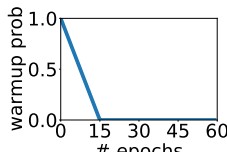
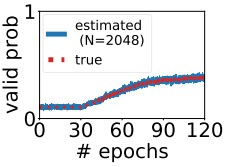

(a) Warmup probability    (b) Valid probability

Figure 4: Illustration of weight-sharing on two-layer FFNs for a binary classification task. Edges denote weights; arrows at the end of lines denote ReLU activations; circles denote hidden nodes; the square in the output layer denotes the output logit. The size of each hidden layer can be one of $\{2, 3, 4\}$, thus the SuperNet is a two-layer FFN with size 4-4. At this moment, the controller picks the child network 3-2, thus only the first 3 hidden nodes in the first hidden layer and the first 2 hidden nodes in the second hidden layer, together with the connected edges (in red), are enabled to compute the output logits.

Figure 5: Examples of layer warmup and valid probabilities. Figure (a) shows our schedule: linearly decay from 1 to 0 in the first 25% epochs. Figure (b) shows an example of the change of true and estimated valid probabilities ($\mathbb{P}(V)$ and $\widehat{\mathbb{P}}(V)$) in a successful search, with 8,000 architectures in the search space and the number of MC samples $N = 1024$. Both probabilities are (nearly) constant during warmup before RL starts, then increase after RL starts because of rejection sampling.

- On the other hand, the factorized search space learns a separate (independent) probability distribution for the choices of each layer. While this distribution is efficient to learn, independence between layers discourages an RL controller with a resource-aware reward from choosing a bottleneck structure. A bottleneck requires the controller to select large sizes for some layers and small for others. But decisions for different layers are made independently, and both very large and very small layer sizes, considered independently, have poor expected rewards: small layers are estimated to perform poorly, while large layers easily exceed the resource constraints.

To bridge the gap and efficiently learn layerwise distributions that take into account the architecture feasibility, we propose a rejection-based RL reward for Algorithm 1. We next sketch the idea; detailed pseudocode is provided as Algorithm 2 in Appendix B.

REINFORCE optimizes a set of logits $\{\ell_{ij}\}_{i\in[L],j\in[C_i]}$ which define a probability distribution $p$ over architectures. In the original algorithm, we sample a random architecture $y$ from $p$ and then estimate its quality $Q(y)$. Updates to the logits $\ell_{ij}$ take the form $\ell_{ij} \leftarrow \ell_{ij} + \eta\frac{\partial}{\partial\ell_{ij}}J(y)$, where $\eta$ is the learning rate, $\overline{Q}$ is a moving average of recent rewards, and $J(y) = \text{stop\_grad}(Q(y) - \overline{Q}) \cdot \log\mathbb{P}(y)$. If $y$ is better (worse) than average, then $Q(y) - \overline{Q}$ will be positive (negative), so the REINFORCE update will increase (decrease) the probability of sampling the same architecture in the future.

In our new REINFORCE variant, motivated by rejection sampling, we do not update the logits when $y$ is infeasible. When $y$ is feasible, we replace the probability $\mathbb{P}(y)$ in the REINFORCE update equation with the conditional probability $\mathbb{P}(y \mid y \in V) = \mathbb{P}(y)/\mathbb{P}(y \in V)$. So $J(y)$ becomes

$$J(y) = \text{stop\_grad}(Q(y) - \overline{Q}) \cdot \log\left[\mathbb{P}(y)/\mathbb{P}(y \in V)\right]. \tag{1}$$

We can compute the probability of sampling a feasible architecture $\mathbb{P}(V) := \mathbb{P}(y \in V)$ exactly when the search space is small, but this computation is too expensive when the space is large. Instead, we replace the exact probability $\mathbb{P}(y)$ with a differential approximation $\widehat{\mathbb{P}}(y)$ obtained with Monte-Carlo (MC) sampling. In each RL step, we sample $N$ architectures $\{z^{(k)}\}_{k\in[N]}$ within the search space with a proposal distribution $q$ and estimate $\mathbb{P}(V)$ as

$$\widehat{\mathbb{P}}(V) = \frac{1}{N}\sum_{k\in[N]}\frac{p^{(k)}}{q^{(k)}} \cdot \mathbb{1}(z^{(k)} \in V). \tag{2}$$

For each $k \in [N]$, $p^{(k)}$ is the probability of sampling $z^{(k)}$ with the factorized layerwise distributions and so is differentiable with respect to the logits. In contrast, $q^{(k)}$ is the probability of sampling $z^{(k)}$ with the proposal distribution, and is therefore non-differentiable.

$\widehat{\mathbb{P}}(V)$ is an unbiased and consistent estimate of $\mathbb{P}(V)$; $\nabla\log[\mathbb{P}(y)/\widehat{\mathbb{P}}(V)]$ is a consistent estimate of $\nabla\log[\mathbb{P}(y \mid y \in V)]$ (Appendix J). A larger $N$ gives better results (Appendix H); in experiments, we

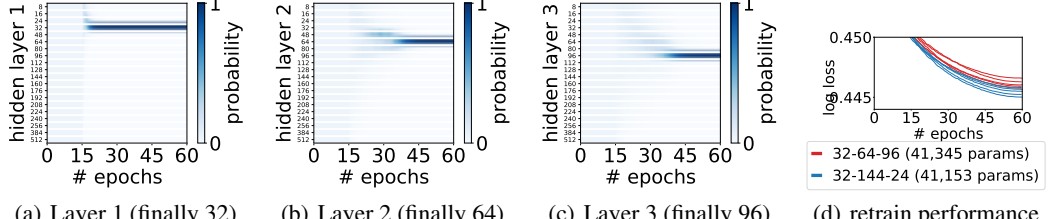

(a) Layer 1 (finally 32)    (b) Layer 2 (finally 64)    (c) Layer 3 (finally 96)    (d) retrain performance

Figure 6: **Failure case of the Abs Reward on Criteo** in a search space of 3-layer FFNs. The change of sampling probabilities and comparison of retrain performance between the 32-144-24 reference and the 32-64-96 architecture found with the $Q(y) + \beta|T(y)/T_0 - 1|$ Abs Reward, the target for the reward was 41,153 parameters. Repeated runs of the same search find the same architecture. Figure 6(d) shows the retrain validation losses of 32-64-96 (NAS-found) and 32-144-24 (reference).

need smaller than the size of the sample space to get a faithful estimate (Figure 5(b), Appendix D and I) because neighboring RL steps can correct the estimates of each other. We set $q = \text{stop\_grad}(p)$ in experiments for convenience: use the current distribution over architectures for MC sampling. Other distributions that have a larger support on $V$ may be used to reduce sampling variance (Appendix J).

At the end of NAS, we pick as our final architecture the layer sizes with largest sampling probabilities if the layerwise distributions are deterministic, or sample from the distributions $m$ times and pick $n$ feasible architectures with the largest number of parameters if not. Appendix B Algorithm 3 provides the full details. We find $m = 500$ and $n \leq 3$ suffice to find an architecture that matches the reference (optimal) architecture in our experiments.

In practice, the distributions often (almost) converge after twice the number of epochs used to train a stand-alone child network. Indeed the distributions are often useful after training the same number of epochs in that the architectures found by Algorithm 3 are competitive. Figure 1 shows TabNAS finds the best feasible architecture, 4-2, in our toy example, using $\widehat{\mathbb{P}}(V)$ estimated by MC sampling.

## 4    Experimental results

Our implementation can be found at `https://github.com/google-research/tabnas`. We ran all experiments using TensorFlow on a Cloud TPU v2 with 8 cores. We use a 1,027-dimensional input representation for the Criteo dataset and 180 features for Volkert[1]. The best architectures in our FFN search spaces already produce near-state-of-the-art results; details in Appendix C.2. More details of experiment setup and results in other search spaces can be found in Appendix C and D. Appendix E tabulates the performance of all RL rewards on all tabular datasets in our experiments. Appendix F shows a comparison with Bayesian optimization and evolutionary search in similar settings; Ablation studies in Appendix I show TabNAS components collectively deliver desirable results; Appendix H shows TabNAS has easy-to-tune hyperparameters.

### 4.1    When do previous RL rewards fail?

Section 3.3 discussed the resource-aware RL rewards and highlighted a potential failure case. In this section, we show several failure cases of three resource-aware rewards, $Q(y)(T(y)/T_0)^\beta$, $Q(y) \max\{1, (T(y)/T_0)^\beta\}$, and the Abs Reward $Q(y) + \beta|T(y)/T_0 - 1|$, on our tabular datasets.

#### 4.1.1    Criteo – 3 layer search space

We use the 32-144-24 reference architecture (41,153 parameters). Figure 3 gives an overview of the costs and losses of all architectures in the search space. The search space requires us to choose one of 20 possible sizes for each hidden layer; details in Appendix D. The search has $1.7\times$ the cost of a stand-alone training run.

---

[1]Our paper takes these features as given. It is worth noting that methods proposed in feature engineering works like [12] and [13] are complementary to and can work together with TabNAS.

**Failure of latency rewards.** Figure 6 shows the sampling probabilities from the search when using the Abs Reward, and the retrain validation losses of the found architecture 32-64-96. In Figures 6(a) – 6(c), the sampling probabilities for the different choices are uniform during warmup and then converge quickly. The final selected model (32-64-96) is much worse than the reference model (32-144-24) even though the reference model is actually less expensive. We also observed similar failures for the MnasNet rewards. With the MnasNet rewards, the RL controller also struggles to find a model within $\pm 5\%$ of the constraint despite a grid search of the RL parameters (details in Appendix C). In both cases, almost all found models are worse than the reference architecture.

**The RL controller is to blame.** To verify that a low quality SuperNet was not the culprit, we trained a SuperNet without updating the RL controller, and manually inspected the quality of the resulting SuperNet. The sampling probabilities for the RL controller remained uniform throughout the search; the rest of the training setup was kept the same. At the end of the training, we compare two sets of losses on each of the child networks: the validation loss from the SuperNet (*one-shot loss*), and the validation loss from training the child network from scratch. Figure 7(a) shows that there is a strong correlation between these accuracies; Figure 7(b) shows RL that starts from the sufficiently trained SuperNet weights in 7(a) still chooses the suboptimal choice 64. This suggests that the suboptimal search results on Criteo are likely due to issues with the RL controller, rather than issues with the one-shot model weights. In a 3 layer search space we can actually find good models without the RL controller, but in a 5 layer search space, we found an RL controller whose training is interleaved with the SuperNet is important to achieve good results.

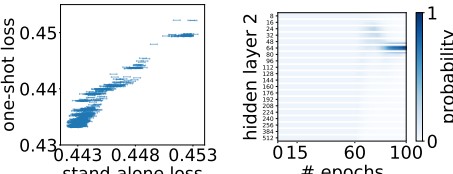

Figure 7: Left: 3-layer Criteo SuperNet calibration after 60 epochs (search space in Appendix C): Pearson correlation is 0.96. The one-shot loss is validation loss of each child network with weights taken from a SuperNet trained with the same hyperparameters as in Figure 6 but with no RL in the first 60 epochs; the stand-alone loss of each child network is computed by training the same architecture with the same hyperparameters from scratch, and has std 0.0003. Right: change in probabilities in layer 2 after 60 epochs of SuperNet training and 40 of RL. Note the rapid changes due to RL.

#### 4.1.2 Volkert – 4 layer search space

We search for 4-layer and 9-layer networks on the Volkert dataset; details in Appendix D. For resource-aware RL rewards, we ran a grid search over the RL learning rate and $\beta$ hyperparameter. The reference architecture for the 4 layer search space is 48-160-32-144 with 27,882 parameters. Despite a hyperparameter grid search, it was difficult to find models with the right target cost reliably using the MnasNet rewards. Using the Abs Reward (Figure 8), searched models met the target cost but their quality was suboptimal, and the trend is similar to what has been shown in the toy example (Figure 1): a smaller $|\beta|$ gives an infeasible architecture that is beyond the reference number of parameters, and a larger $|\beta|$ gives an architecture that is feasible but suboptimal.

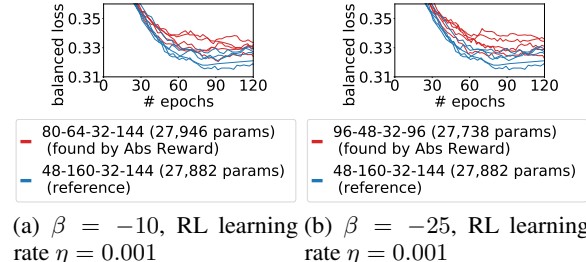

(a) $\beta = -10$, RL learning rate $\eta = 0.001$

(b) $\beta = -25$, RL learning rate $\eta = 0.001$

Figure 8: Abs Reward misses the global optimum on Volkert. Figure shows the retrain validation losses of two architectures found by the Abs Reward vs. the 48-160-32-144 reference.

#### 4.1.3 A common failure pattern

Apart from Section 4.1.1 and 4.1.2, more examples in search spaces of deeper FFNs can be found in Appendix D. In cases on Criteo and Volkert where where the RL controller with soft constraints cannot match the quality of the reference architectures, the reference architecture often has a bottleneck structure. For example, with a 1,027-dimensional input representation, the 32-144-24 reference on Criteo has bottleneck 32; with 180 features, the 48-160-32-144 reference on Volkert has bottleneck 48 and 32. As the example in Section 3.3 shows, the wide hidden layers around the bottlenecks get penalized harder in the search, and it is thus more difficult for RL with the Abs Reward to find a

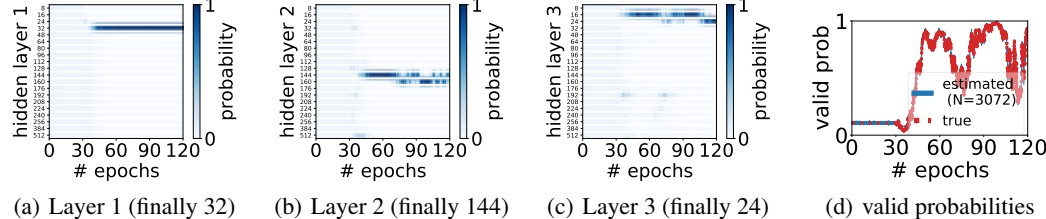

(a) Layer 1 (finally 32)     (b) Layer 2 (finally 144)     (c) Layer 3 (finally 24)     (d) valid probabilities

Figure 9: **Success case:** on Criteo in a search space of 3-layer FFNs, Monte-Carlo sampling with rejection eventually finds 32-144-24, the reference architecture, with RL learning rate 0.005 and number of MC samples 3,072. Figure 9(d) shows the change of true and estimated valid probabilities.

model that can match the reference performance. Also, Appendix C.2.1 shows the Pareto-optimal architectures in the tradeoff points in Figure 3 often have bottleneck structures, so resource-aware RL rewards in previous NAS practice may have more room for improvement than previously believed.

### 4.2 NAS with TabNAS reward

With proper hyperparameters (Appendix H), our RL controller with TabNAS reward finds the global optimum when RL with resource-aware rewards produces suboptimal results.

TabNAS does not introduce a resource-aware bias in the RL reward (Section 3.3). Instead, it uses conditional probabilities to update the logits in feasible architectures. We run TabNAS for 120 epochs with RL learning rate 0.005 and $N = 3072$ MC samples.[2] The RL controller converges to two architectures, 32-160-16 (40,769 parameters, with loss $0.4457 \pm 0.0002$) and 32-144-24 (41,153 parameters, with loss $0.4455 \pm 0.0003$), after around 50 epochs of NAS, then oscillates between these two solutions (Figure 9). After 120-epochs, we sample from the layerwise distribution and pick the largest feasible architecture: the global optimum 32-144-24.

On the same hardware, the search takes $3\times$ the runtime of stand-alone training. Hence, as can be seen in Figure 2, the proposed architecture search method is much faster than a random baseline.

### 4.3 TabNAS automatically determines whether bottlenecks are needed

Previous NAS works like MnasNet and TuNAS (often or only on vision tasks) often have inverted bottleneck blocks [17] in their search spaces. However, the search spaces used there have a hard-coded requirement that certain layers must have bottlenecks. In contrast, our search spaces permit the controller to *automatically determine* whether to use bottleneck structures based on the task under consideration. TabNAS automatically finds high-quality architectures, both in cases where bottlenecks are needed and in cases where they are not. This is important because networks with bottlenecks do not always outperform others on all tasks. For example, the reference architecture 32-144-24 outperforms the TuNAS-found 32-64-96 on Criteo, but the reference 64-192-48-32 (64,568 parameters, $0.0662 \pm 0.0011$) is on par with the TuNAS-and-TabNAS-found 96-80-96-32 (64,024 parameters, $0.0669 \pm 0.0013$) on Aloi. TabNAS automatically finds an optimal (bottleneck) architecture for Criteo, and automatically finds an optimal architecture that does not necessarily have a bottleneck structure for Aloi. Previous reward-shaping rewards like the Abs Reward only succeed in the latter case.

### 4.4 Rejection-based reward outperforms Abs Reward in NATS-Bench size search space

Although we target resource-constrained NAS on tabular datasets in this paper, our proposed method is not specific to NAS on tabular datasets. In Appendix G, we show the rejection-based reward in TabNAS outperforms RL with the Abs Reward in the size search space of NATS-Bench [6], a NAS benchmark on vision tasks.

---

[2]The 3-layer search space has $20^3 = 8000$ candidate architectures, which is small enough to compute $\mathbb{P}(V)$ exactly. However, MC can scale to larger spaces which are prohibitively expensive for exhaustive search (Appendix D).

# 5 Conclusion

We investigate the failure of resource-aware RL rewards to discover optimal structures in tabular NAS and propose TabNAS for tabular NAS in a constrained search space. The TabNAS controller uses a rejection mechanism to compute the policy gradient updates from feasible architectures only, and uses Monte-Carlo sampling to reduce the cost of debiasing this rejection-sampling approach. Experiments show TabNAS finds better architectures than previously proposed RL methods with resource-aware rewards in resource-constrained searches.

Many questions remain open. For example: 1) Can the TabNAS strategy find better architectures on other types of tasks such as vision and language? 2) Can TabNAS improve RL results for more complex architectures? 3) Is TabNAS useful for resource-constrained RL problems more broadly?

## Acknowledgments and Disclosure of Funding

This work was done when Madeleine Udell was a visiting researcher at Google. The authors thank Ruoxi Wang, Mike Van Ness, Ziteng Sun, Xuanyi Dong, Lijun Ding, Yanqi Zhou, Chen Liang, Zachary Frangella, Yi Su, and Ed H. Chi for helpful discussions, and thank several anonymous reviewers for useful comments.

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
