# Appendix for TabNAS: Rejection Sampling for Neural Architecture Search on Tabular Datasets

**Chengrun Yang**[1], **Gabriel Bender**[1], **Hanxiao Liu**[1], **Pieter-Jan Kindermans**[1],
**Madeleine Udell**[2], **Yifeng Lu**[1], **Quoc V. Le**[1], **Da Huang**[1]
{chengrun, gbender, hanxiaol, pikinder}@google.com,
udell@stanford.edu, {yifenglu, qvl, dahua}@google.com

[1] Google Research, Brain Team  [2] Stanford University

## A    Previous work

### A.1    Neural architecture search (NAS)

Neural architecture search (NAS) [52] stems from the resurgence of deep learning. It focuses on tuning architectural hyperparameters in neural networks, like hidden layer sizes in feedforward networks or convolutional kernel sizes in convolutional networks. The earliest NAS papers [52, 53] trained thousands of network architectures from scratch for a single search. Due to the high costs involved, many works have proposed different methods to reduce the search cost. Most of these proposals are based around two complementary (but often intertwined) high-level strategies.

The first strategy is to reduce the time needed to evaluate each architecture seen during a search. For example, instead of training each network architecture from scratch, we can train a SuperNet – a single set of shared model weights that can be used to evaluate and rank any different candidate architecture in the search space [5, 29, 37, 49]. Other approaches include the use of network morphism [24, 12] to initialize weights of candidate networks that are close to previous instances.

The second strategy is to reduce the number of architectures we need to evaluate during a search. Proposed methods include reinforcement learning algorithms [52, 7, 6] that learn a probability distribution over candidate architectures, evolutionary search [28, 12, 18, 3, 36] that pursues more promising architectures from ancestors, and Bayesian optimization [24, 21, 43, 50, 37] and parametric models [48, 42] that directly predict network performance.

Resource constraints are prevalent in deep learning. Finding architectures with outstanding performance and low costs are important to both NAS research and application. Apart from the surrogate models above that transfer knowledge across network candidates to avoid exhaustive search, specific techniques have been adopted to find networks with a good balance of performance and resource consumption. A popular method is to add regularizers [44, 39, 6, 30] that penalize expensive architectures. With hard resource constraints, greedy submodular maximization [45] and heuristic scaling methods [19] were used to grow or shrink networks during the search, so as to ensure the chosen candidate architecture obeys the constraint. TabNAS in this work operates under the hard resource constraint, and can find the global optima in the feasible set of architectures.

### A.2    Deep learning on tabular datasets

Deep neural networks are gaining popularity on tabular datasets in academia and industry. In the pursuit of designing better architectures for tabular deep learning, an earlier line of work [26, 2, 34, 1, 8] mimic the structure of tree-based models [9, 25]. Some other works use the attention mechanism [20, 38, 16] or are based on feedforward [23, 16] or residual networks [16].

While automated machine learning (AutoML) on tabular datasets has been addressed from multiple perspectives and with different approaches [14, 33, 15, 46, 13], NAS on tabular datasets is less ex-

plored, partly due to insufficient understanding of promising architectures and a lack of benchmarks [1]. Egele et al. [11] interleaves NAS with aging evolution [36] in a search space with multiple branches and hyperparameter tuning with Bayesian optimization. We show TabNAS can find architectures as simple as FFNs with a few layers that have outstanding performance and obey the resource constraint.

# B   Algorithm pseudocode

We show pseudocode of the algorithms introduced in Section 3.

---

**Algorithm 1** (Resource-Oblivious) One-Shot Training and REINFORCE

---

**Input:**  search space $S$, weight learning rate $\alpha$, RL learning rate $\eta$
**Output:**  sampling probabilities $\{p_{ij}\}_{i\in[L],j\in[C_i]}$

1   initialize logits $\ell_{ij} \leftarrow 0, \forall i \in [L], j \in [C_i]$
2   initialize quality reward moving average $\bar{Q} \leftarrow 0$
3   layer warmup
4   **for** iter $= 1$ **to** max_iter **do**
5       $p_{ij} \leftarrow \exp(\ell_{ij})/\sum_{j\in[C_i]} \exp(\ell_{ij}), \forall i \in [L], j \in [C_i]$
6                                                                     ▷ weight update
7       **for** $i = 1$ to $L$ **do**
8           $x_i \leftarrow$ the $i$-th layer size sampled from $\{s_{ij}\}_{j\in[C_i]}$ with distribution $\{p_{ij}\}_{j\in[C_i]}$
9       $loss(x) \leftarrow$ the (training) loss of $x = x_1\text{-}\cdots\text{-}x_L$ on the training set
10      $w \leftarrow w - \alpha\nabla loss(x)$, in which $w$ is the weights of $x$       ▷ can be replaced with optimizers other than SGD
11                                                                     ▷ RL update
12      **for** $i = 1$ to $L$ **do**
13          $y_i \leftarrow$ the $i$-th layer size sampled from $\{s_{ij}\}_{j\in[C_i]}$ with distribution $\{p_{ij}\}_{j\in[C_i]}$
14      $Q(y) \leftarrow 1 - loss(y)$, the quality reward of $y = y_1\text{-}\cdots\text{-}y_L$ on the validation set
15      RL reward $r(y) \leftarrow Q(y)$        ▷ can be replaced with resource-aware rewards introduced in Section 3.3
16      $J(y) \leftarrow \text{stop\_grad}(r(y) - \bar{Q}) \log \mathbb{P}(y)$              ▷ can be replaced with Algorithm 2 when resource-constrained
17      $\ell_{ij} \leftarrow \ell_{ij} + \eta\nabla J(y), \forall i \in [L], j \in [C_i]$  ▷ can be replaced with optimizers other than SGD
18      $\bar{Q} \leftarrow \frac{\gamma*\bar{Q}+(1-\gamma)*Q(y)}{\gamma*\bar{Q}+1-\gamma}$                          ▷ update moving average with $\gamma = 0.9$

---

[1]To disambiguate, the phrase "tabular NAS benchmark" in previous NAS benchmark literature [47, 31] often refers to tabulated performance of architectures on vision and language tasks.

---

**Algorithm 2** Rejection with Monte-Carlo (MC) Sampling

---

1  **Input:** number of MC samples $N$, feasible set $V$, MC proposal distribution $q$, quality reward moving average $\bar{Q}$, sampled architecture for RL in the current step $y = y_1\text{-}y_2\text{-}\cdots\text{-}y_L$, current layer size distribution over $\{s_{ij}\}_{j\in[C_i]}$ with probability $\{p_{ij}\}_{j\in[C_i]}$

2  **Output:** $J(y)$

3  **if** $y$ is feasible **then**

4      $Q(y) = $ the quality reward of $y$

5      $\mathbb{P}(y) := \prod\limits_{i\in[L]} \mathbb{P}(Y_i = y_i)$

6      **for** $i = 1$ to $L$ **do**

7          $\{z_i^{(k)}\}_{k\in[N]} \leftarrow N$ samples of the $i$-th layer size, sampled from $\{s_{ij}\}_{j\in[C_i]}$ with distribution $\{p_{ij}\}_{j\in[C_i]}$

8          $p_i^{(k)} := \mathbb{P}(Z_i = z_i^{(k)}), \forall i \in [L], k \in [N]$

9          $p^{(k)} := \prod\limits_{i\in[L]} p_i^{(k)}, \forall k \in [N]$

10     $\widehat{\mathbb{P}}(V) \leftarrow \frac{1}{N} \sum\limits_{k\in[N], z^{(k)}\in V} \frac{p^{(k)}}{q^{(k)}}$, in which $z^{(k)} := z_1^{(k)}\text{-}\cdots\text{-}z_L^{(k)}$

11     $J(y) \leftarrow \text{stop\_grad}(Q(y) - \bar{Q}) \log \frac{\mathbb{P}(y)}{\widehat{\mathbb{P}}(V)}$

12 **else**

13     $J(y) \leftarrow 0$

---

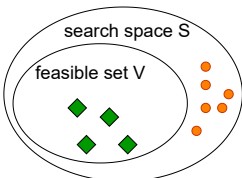

Figure A1: Illustration of the feasible set $V$ within the search space $S$. Each green diamond or orange dot denotes a feasible or infeasible architecture, respectively.

---

**Algorithm 3** Sample to Return the Final Architecture

---

1  **Input:** sampling probabilities $\{p_{ij}\}_{i\in[L], j\in[C_i]}$ returned by Algorithm 1, number of desired architectures $n$, number of samples to draw $m$

2  **Output:** the set of $n$ selected architectures $A$

3  **for** $i = 1$ to $L$ **do**

4      $\{x_i^{(k)}\}_{k\in[m]} \leftarrow m$ samples of the $i$-th layer size, sampled from $\{s_{ij}\}_{j\in[C_i]}$ with distribution $\{p_{ij}\}_{j\in[C_i]}$

5  $F := \{k \in [m] \mid x_1^{(k)}\text{-}x_2^{(k)}\text{-}\cdots\text{-}x_L^{(k)} \in V\}$

6  $A \leftarrow n$ unique architectures in $F$ with largest numbers of parameters

---

Notice that in Algorithm 1, we show the weight and RL updates with the stochastic gradient descent (SGD) algorithm; in our experiments on the toy example and real datasets, we use Adam for both updates as in ProxylessNAS [7] and TuNAS [6], since it synchronizes convergence across different layer size choices, and slows down the learning which would otherwise converge too rapidly.

## C  Details of experiment setup

### C.1  Toy example

We use the Adam optimizer with $\beta_1 = 0.9$, $\beta_2 = 0.999$ and $\epsilon = 0.001$ to update the logits. When we use the Abs Reward, the results are similar when $\eta \geq 0.05$, while the RL controller with $\eta < 0.05$ converges too slow or is hard to converge. When we use the rejection-based reward, we use RL learning rate $\eta = 0.1$; other $\eta$ values with which RL converges give similar results.

## C.2 Real datasets

Table A1 shows the datasets we use. Datasets other than Criteo [2] come from the OpenML dataset repository [40]. For Criteo, we randomly split the labeled part (45,840,617 points) into 90% training (41,258,185 points) and 10% validation (4,582,432 points); for the other datasets, we randomly split into 80% training and 20% validation[3]. The representations we use for Criteo are inspired by DCN-V2 [41].

Table A1: Dataset details

| name | # points | # features | | # classes | embedding we use for each feature |
| | | numerical | categorical | | |
|---|---|---|---|---|---|
| Criteo | 51,882,752 | 13 | 26 | 2 | original values for each numerical, 39-dimensional for each categorical |
| Volkert | 58,310 | 180 | 0 | 10 | original values |
| Aloi | 108,000 | 128 | 0 | 1,000 | original values |
| Connect-4 | 67,557 | 0 | 42 | 3 | 2-dimensional for each categorical |
| Higgs | 98,050 | 28 | 0 | 2 | original values |

Table A2 shows the hyperparameters we use for stand-alone training and NAS, found by grid search. With these hyperparameters, the best architecture in each of our search spaces (introduced in Appendix C.2.1) has performance that is within $\pm 5\%$ of the best performance in Kadra et al. [23] Table 2, and we achieve these scores with FFNs that only have 5% parameters of the ones there. The Adam optimizer has hyperparameters $\beta_1 = 0.9$, $\beta_2 = 0.999$ and $\epsilon = 0.001$. We use layer normalization [4] for all datasets. We use balanced error (weighted average of classification errors across classes) for all other datasets[4] as in Kadra et al. [23], except for Criteo, on which we use logistic loss as in Wang et al. [41].

Table A2: Weight training hyperparameter details

| name | batch size | learning rate | learning rate schedule | optimizer | # training epochs | metric |
|---|---|---|---|---|---|---|
| Criteo | 512 | 0.002 | cosine decay | Adam | 60 | log loss |
| Volkert | 32 | 0.01 | constant | SGD with momentum 0.9 | 120 | balanced error |
| Aloi | 128 | 0.0005 | constant | Adam | 50 | balanced error |
| Connect-4 | 32 | 0.0005 | cosine decay | Adam | 60 | balanced error |
| Higgs | 64 | 0.05 | constant | SGD | 60 | balanced error |

We use constant RL learning rates for NAS. The Connect-4[5] and Higgs[6] datasets are easy for both the Abs Reward and rejection-based reward, in the sense that small FFNs with fewer than 5,000 parameters can achieve near-SOTA results ($\pm 5\%$ of the best accuracy scores listed in Kadra et al. [23] Table 2, except that we do 80%-20% training-validation splits and use original instead of standardized features), and RL-based weight-sharing NAS with either reward can find architectures that match the Pareto-optimal reference architectures. The Aloi dataset[7] needs more parameters (more than 100k),

---

[2]https://ailab.criteo.com/download-criteo-1tb-click-logs-dataset/

[3]The ranking of validation losses among architectures under such splits is almost the same as that of test losses under 60%-20%-20% training-validation-test splits.

[4]The performance ranking of architectures under the balanced error metric is almost the same as under logistic loss. Also, the balanced error metric is only for reporting the final validation losses; both weight and RL updates use logistic loss.

[5]https://www.openml.org/d/40668

[6]https://www.openml.org/d/23512

[7]https://www.openml.org/d/42396

but the other observations are similar to on Connect-4 and Higgs. Thus we omit the corresponding results.

The factorized search spaces we use for NAS are:

- Criteo: Each layer has 20 choices {8, 16, 24, 32, 48, 64, 80, 96, 112, 128, 144, 160, 176, 192, 208, 224, 240, 256, 384, 512}.
- Volkert[8], 4-layer networks: Each layer has 20 choices {8, 16, 24, 32, 48, 64, 80, 96, 112, 128, 144, 160, 176, 192, 208, 224, 240, 256, 384, 512}.
- Volkert, 9-layer networks: Each layer has 12 choices {8, 16, 24, 32, 48, 64, 80, 96, 112, 128, 144, 160}. This search space has fewer choices for each hidden layer than the 4-layer counterpart, but the size of the search space is over $3 \times 10^4$ times larger.

Our goal is not to achieve state-of-the-art (SOTA) accuracy, but to find the best architecture that obeys a resource upper bound. This mimics a resource-constrained setting that is common in practice. Impressively, our method does nearly match SOTA performance, as our search space has architectures that are close to the best in previous literature. For example:

- On Criteo: The best architecture in our search space achieves public and private scores 0.45284 and 0.45283 on Kaggle, ranking 20/717 on the leaderboard[9].
- On Volkert: The best architecture in our search space has balanced accuracy 0.695, within 2% of the best in Kadra et al. [23] and better than most other works used for comparison in that work. The differences in settings are that we use original features instead of standardized, and we achieve this score with an FFN that only has 5% parameters of the one there.
- On Aloi: The best architecture in our search space has balanced accuracy 0.957, within 2% of the best in Kadra et al. [23] and better than most other works used for comparison in that work. Again, we use original features instead of standardized, and we achieve this score with an FFN that only has 5% parameters of the one used there. And the 0.957 balanced accuracy score is also within 1% of the best in Gorishniy et al. [16]. The difference is that we use an FFN that only has <10% as many parameters as the one used there.

### C.2.1 More details on the tradeoff plot (Figure 3)

Each search space we use for exhaustive search and NAS has a fixed number of hidden layers. Resource-constrained NAS in a search space with varying number of hidden layers is an interesting problem for future studies. On each dataset, we randomly sample, train and evaluate architectures in the search space with the number of parameters fall within a range, in which there is a clear tradeoff between loss and number of parameters. These ranges are:

- Criteo: 0 – 200,000
- Volkert, 4-layer networks: 15,000 – 50,000
- Volkert, 9-layer networks: 40,000 – 100,000

Figure A2 shows the tradeoffs between loss and number of parameters in these search spaces. When training each architecture 5 times, the standard deviation (std) across different runs is 0.0002 for Criteo[10] and 0.004 for Volkert, meaning that the architectures whose performance difference is larger than $2\times$ std are qualitatively different. We use Pareto-optimal architectures as the reference of resource-constrained NAS: we want an architecture that both matches (or even beats[11]) the performance of the reference architecture and has no more parameters than the reference. Most Pareto-optimal architectures in Figure A2 have the bottleneck structure; Table A3 shows some examples.

---

[8]https://www.openml.org/d/41166

[9]https://www.kaggle.com/competitions/criteo-display-ad-challenge/leaderboard

[10]On Criteo, "a 0.001-level improvement (of logistic loss) is considered significant" [41].

[11]Note that the Pareto optimality of the reference architecture is determined by only one round of random search. Thus because of the randomness across multiple training runs, the other architectures are likely to beat the reference architecture: a "regression toward the mean".

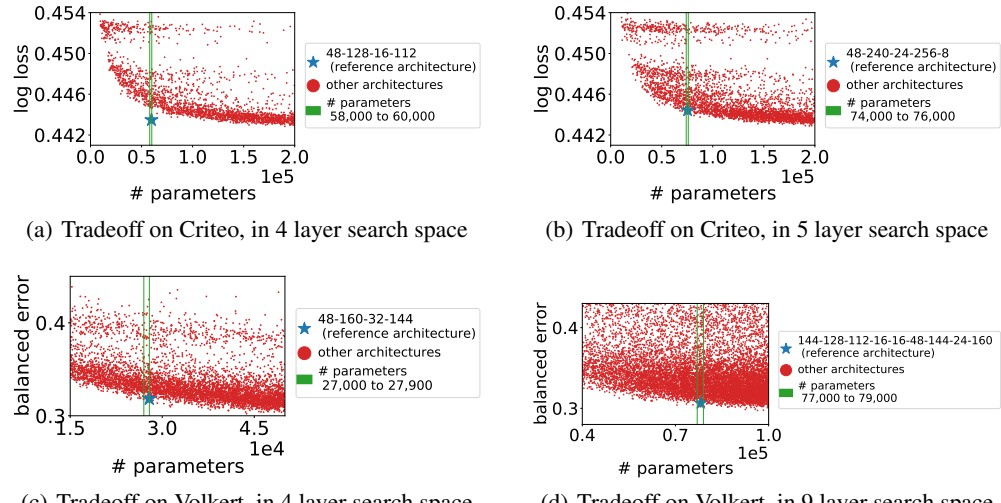

(a) Tradeoff on Criteo, in 4 layer search space      (b) Tradeoff on Criteo, in 5 layer search space

(c) Tradeoff on Volkert, in 4 layer search space      (d) Tradeoff on Volkert, in 9 layer search space

Figure A2: Tradeoffs between validation loss and number of parameters in four search spaces.

Table A3: Some Pareto-optimal architectures in Figure A2. All architectures shown here and almost all other Pareto-optimal architectures have the bottleneck structure.

|  | search space | Pareto-optimal architecture | number of parameters | loss |
|---|---|---|---|---|
| Figure A2(a) | Criteo 4-layer | 32-144-24-112 | 44,041 | 0.4454 |
| Figure A2(a) | Criteo 4-layer | 48-112-8-80 | 56,537 | 0.4448 |
| Figure A2(a) | Criteo 4-layer | 48-384-16-176 | 77,489 | 0.4441 |
| Figure A2(a) | Criteo 4-layer | 96-144-32-240 | 125,457 | 0.4433 |
| Figure A2(a) | Criteo 4-layer | 96-384-48-16 | 155,217 | 0.4430 |
| Figure A2(b) | Criteo 5-layer | 32-240-16-8-96 | 45,769 | 0.4451 |
| Figure A2(b) | Criteo 5-layer | 48-128-64-16-128 | 67,217 | 0.4446 |
| Figure A2(b) | Criteo 5-layer | 48-256-16-8-384 | 69,977 | 0.4443 |
| Figure A2(b) | Criteo 5-layer | 64-144-48-96-160 | 102,497 | 0.4437 |
| Figure A2(b) | Criteo 5-layer | 96-512-24-256-48 | 179,449 | 0.4430 |
| Figure A2(c) | Volkert 4-layer | 48-112-16-24 | 16,642 | 0.3314 |
| Figure A2(c) | Volkert 4-layer | 32-112-24-224 | 20,050 | 0.3269 |
| Figure A2(c) | Volkert 4-layer | 48-160-32-144 | 27,882 | 0.3149 |
| Figure A2(c) | Volkert 4-layer | 48-256-24-112 | 31,330 | 0.3097 |
| Figure A2(c) | Volkert 4-layer | 80-208-32-64 | 40,778 | 0.3054 |
| Figure A2(d) | Volkert 9-layer | 64-64-160-48-16-144-16-8-48 | 40,482 | 0.3250 |
| Figure A2(d) | Volkert 9-layer | 80-144-32-112-32-8-128-8-144 | 43,290 | 0.3238 |
| Figure A2(d) | Volkert 9-layer | 112-144-32-32-24-24-24-128-32 | 51,890 | 0.3128 |
| Figure A2(d) | Volkert 9-layer | 144-128-112-16-16-48-144-24-160 | 78,114 | 0.3019 |
| Figure A2(d) | Volkert 9-layer | 160-144-144-32-112-32-48-32-144 | 94,330 | 0.3010 |

### C.2.2 More details on TPU implementation

When we run one-shot NAS on a TPU that has multiple TPU cores (for example, each Cloud TPU-v2 we use has 8 cores), each core samples an architectures independently, and we use the average loss and reward for weight and RL updates, respectively. This means our algorithm actually samples multiple architectures in each iteration and uses the `tensorflow.tpu.cross_replica_sum()` method to compute their average effect on the gradient. Since only a fraction of architectures are feasible in each search space, we set the losses and rewards given by the infeasible architectures to 0 before averaging, so that we are equivalently only averaging across the sampled architectures that are feasible. We then reweight the average loss or reward with number_of_cores / number_of_feasible_architectures to obtain an unbiased estimate.

### C.2.3 More details on the NAS method comparison plot (Figure 2)

For each architecture below, we report its number of parameters and mean $\pm$ std logistic loss across 5 stand-alone training runs in brackets.

We have the reference architecture 32-144-24 (41,153 parameters, $0.4454 \pm 0.0003$) for NAS methods to match. In the search space with $20^3 = 8000$ candidate architectures:

- TabNAS trials with no fewer than 2,048 Monte-Carlo samples and the RL learning rate $\eta$ among $\{0.001, 0.005, 0.01\}$ consistently finds either the reference architecture itself, or an architectures that is qualitatively the same as the reference, like 32-112-32 (40,241 parameters, $0.4456 \pm 0.0003$).

- NAS with the Abs Reward: After grid search over RL learning rate $\eta$ (among $\{0.0001, 0.0005, 0.001, 0.005, 0.01, 0.015, 0.02, 0.025, 0.03, 0.04, 0.05, 0.06, 0.07, 0.08, 0.09, 0.1, 0.15, 0.2, 0.25, 0.3, 0.4, 0.5, 0.75, 1.0, 1.5, 2.0\}$) and $\beta$ (among $\{-0.0005, -0.001, -0.005, -0.01, -0.05, -0.1, -0.5, -0.75, -1.0, -1.25, -1.5, -2.0, -3.0\}$), the RL controller finds 32-64-96 (41,345 parameters, $0.4461 \pm 0.0003$) or 32-80-64 (40,785 parameters, $0.4459 \pm 0.0002$) among over $90\%$ trials that eventually find an architecture within $\pm 5\%$ of the target number of parameters 41,153.

### C.3 Difficulty in using the MnasNet reward

With the MnasNet reward, only fewer than 1% NAS trials in our hyperparameter grid search (the ones with a medium $\beta$) can find an architecture whose number of parameters is within $\pm 5\%$ of the reference, and among which none or only one (out of tens) can match the reference performance. In contrast, TuNAS with the Abs Reward finds an architecture with number of parameters within $\pm 5\%$ of the reference among over 50% of the grid search trials described in Appendix C.2.3, and TabNAS with the rejection-based reward consistently finds such architectures at medium RL learning rates $\eta$ and decently large numbers of MC samples $N$. This means it is significantly more difficult to use the MnasNet reward than competing approaches in the practice of resource-constrained tabular NAS.

## D   More failure cases of the Abs Reward, and performance of TabNAS

For each architecture below, we report its number of parameters and mean $\pm$ std loss across 5 stand-alone training runs (logistic loss for Criteo, balanced error for the others) in brackets.

**On Criteo, in the 4-layer search space.** We have the reference architecture 48-128-16-112 (59,697 parameters, $0.4451 \pm 0.0002$) for NAS to match in the search space (shown as Figure A2(a)). Similar to Figure 2, we show similar results on NAS with rejection-based reward (TabNAS) and NAS with the Abs Reward (TuNAS) in Figure A3(a). In the search space with $20^4 = 1.6 \times 10^5$ candidate architectures:

- TabNAS with 32,768 Monte-Carlo samples and RL learning rate $\eta$ among $\{0.001, 0.005, 0.01\}$ consistently finds architectures qualitatively the same as the reference. Example results include 48-128-24-32 (59,545 parameters, $0.4449 \pm 0.0002$), 48-144-16-48 (59,585 parameters, $0.4448 \pm 0.0001$), 48-112-16-144 (59,233 parameters, $0.4448 \pm 0.0002$) and the reference architecture itself.

- NAS with the Abs Reward successfully finds the reference architecture 48-128-16-112 in 3 out of 338 hyperparameter settings on a $\beta$-$\eta$ grid. Other found architectures include 48-80-32-112 (59,665 parameters, $0.4452 \pm 0.0002$), 32-128-80-144 (59,249 parameters, $0.4453 \pm 0.0003$) and 48-160-8-48 (58,953 parameters, $0.4448 \pm 0.0003$), among which the first two are inferior to the TabNAS-found counterparts.

**On Criteo, in the 5-layer search space.** We have the reference architecture 48-240-24-256-8 (75,353 parameters, $0.4448 \pm 0.0002$) for NAS methods to match in the search space (shown as Figure A2(b)). Similar to Figure 2, we have similar results on the comparison among random sampling, NAS with rejection-based reward (TabNAS), and NAS with the Abs Reward as Figure A3(b). In the search space with $20^5 = 3.2 \times 10^6$ candidate architectures:

- TabNAS with 32,768 Monte-Carlo samples and the RL learning rate $\eta = 0.005$ consistently finds architectures qualitatively the same as the reference. Example results include 48-176-64-16-256 (74,945 parameters, $0.4445 \pm 0.0002$), 48-208-48-48-64 (75,121 parameters, $0.4444 \pm$

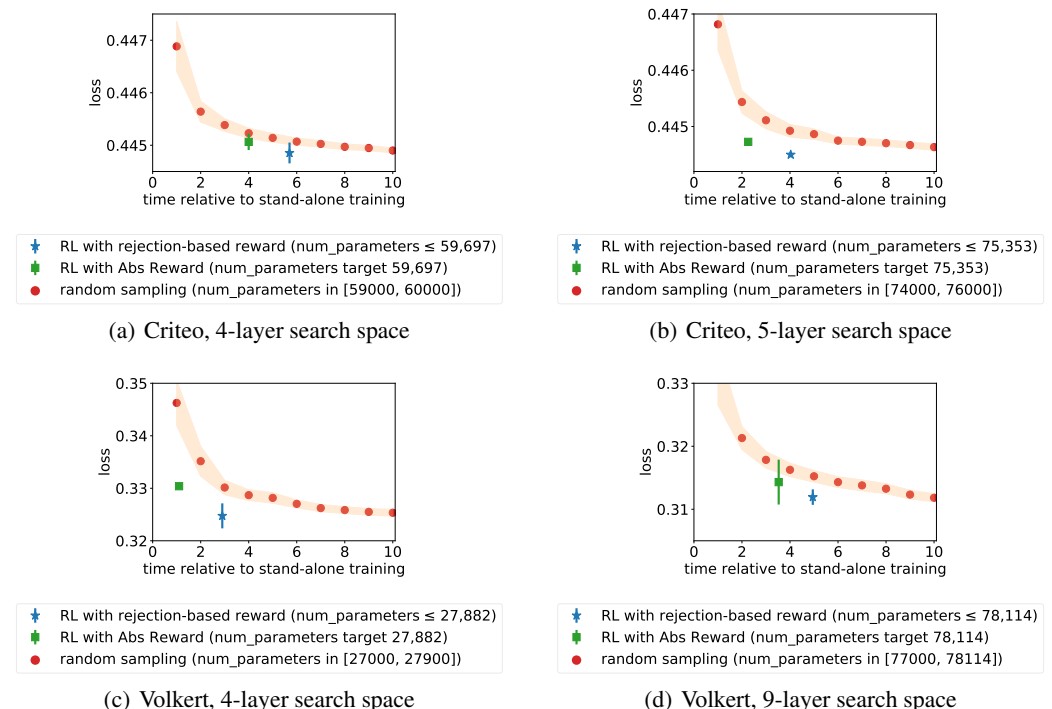

Figure A3: Rejection-based reward distributionally outperforms random search and resource-aware Abs Reward in a number of search spaces. The points and error bars have the same meaning as in Figure 2. The time taken for each stand-alone training run (the unit length for x axes) is 2.5 hours on Criteo (Figure 2 in the main paper and (a), (b)), 10 minutes on Volkert with 4-layer FFNs (Figure (c)), and 22-25 minutes on Volkert with 9-layer FFNs (Figure (d)).

0.0001), 48-256-32-80-24 (74,721 parameters, $0.4446 \pm 0.0003$) and 48-176-80-16-96 (75,153 parameters, $0.4445 \pm 0.0002$).

- NAS with the Abs Reward finds 64-80-48-8-8 (75,353 parameters, $0.4448 \pm 0.0001$), 64-80-24-16-112 (75,353 parameters, $0.4447 \pm 0.0001$), 48-144-96-16-192 (75,329 parameters, $0.4446 \pm 0.0001$) and 64-96-8-32-64 (75,273 parameters, $0.4445 \pm 0.0001$) that are mostly inferior to the TabNAS-found architectures.

**On Volkert, in the 4-layer search space.** We have the reference architecture 48-160-32-144 (27,882 parameters, $0.3244 \pm 0.0040$) for NAS to match in the search space (shown as Figure A2(c)). Similar to Figure 2, we draw the comparison plot among random sampling, NAS with rejection-based reward (TabNAS), and NAS with the Abs Reward as Figure A3(c). In the search space with $1.6 \times 10^5$ candidate architectures:

- TabNAS with $10^4$ Monte-Carlo samples and the RL learning rate $\eta \in \{0.001, 0.005, 0.01, 0.05\}$ consistently finds either the reference architecture itself or other architectures qualitatively the same. Examples include 64-128-48-16 (27,050 parameters, $0.3237 \pm 0.0040$), 80-48-112-32 (27,802 parameters, $0.3274 \pm 0.0037$), 64-96-80-24 (27,778 parameters, $0.3279 \pm 0.0005$), and 64-144-32-48 (27,658 parameters, $0.3204 \pm 0.0038$).
- NAS with the Abs Reward finds 96-64-32-48 (27,738 parameters, $0.3302 \pm 0.0042$), 96-48-32-96 (27,738 parameters, $0.3305 \pm 0.0047$), 96-80-16-48 (27,738 parameters, $0.3302 \pm 0.0050$), 112-48-24-24 (27,722 parameters, $0.3301 \pm 0.0034$) and 80-80-48-48 (27,690 parameters, $0.3309 \pm 0.0022$) that are inferior.

**On Volkert, in the 9-layer search space.** We further do NAS on Volkert in the 9-layer search space to test the ability of TabNAS in searching among significantly deeper FFNs. The tradeoff between loss and number of parameters in the search space is shown in Figure A2(d). We have the reference architecture 144-128-112-16-16-48-144-24-160 (78,114 parameters, $0.3126 \pm 0.0050$) for NAS to match. We compare random sampling, NAS with rejection-based reward (TabNAS), and NAS with

the Abs Reward in Figure A3(d). In the search space with $5.2 \times 10^9$ candidate architectures (which is nearly impossible for exhaustive search):

- TabNAS with $5 \times 10^6$ Monte-Carlo samples and the RL learning rate $\eta \in \{0.002, 0.005\}$ consistently finds architectures that are qualitatively the same as the reference. These architectures are found when the RL controller is far from converged and when $\mathbb{P}(V)$ slightly decreases after RL starts. Example results include 144-144-112-64-24-16-128-8-128 (78,026 parameters, $0.3120 \pm 0.0049$), 128-160-96-32-24-64-64-32-160 (77,890 parameters, $0.3127 \pm 0.0040$), 128-144-112-32-64-64-80-16-128 (77,834 parameters, $0.3094 \pm 0.0012$), 160-128-96-32-48-64-48-24-112 (78,002 parameters, $0.3137 \pm 0.0021$), and 144-112-160-24-112-16-16-128-48 (77,986 parameters, $0.3119 \pm 0.0029$).

- NAS with the Abs Reward finds 144-96-80-80-48-64-96-80-32 (78,170 parameters, $0.3094 \pm 0.0039$), 160-80-160-24-80-16-80-64-128 (78,114 parameters, $0.3158 \pm 0.0020$), 128-96-80-80-64-64-80-80-80 (78,106 parameters, $0.3128 \pm 0.0020$), and 144-128-80-16-16-16-96-160-24 (78,050 parameters, $0.3192 \pm 0.0014$). Interestingly, all architectures except 144-96-80-80-48-64-96-80-32 are inferior to the TabNAS-found architectures despite having slightly more parameters, and 144-96-80-80-48-64-96-80-32 does not have an evident bottleneck structure like the other architectures found here.

## E   Tabulated performance of different RL rewards on all datasets

Table A4 summarizes among results of RL with the rejection-based reward, the Abs Reward, two MNasNet rewards and the reward in RENA [51] Equation 3. Bold results in each column indicate architectures that are on par with the best for the corresponding dataset. The rejection-based reward in TabNAS gets the best architectures overall.

## F   Comparison with Bayesian optimization and evolutionary search in one-shot NAS

Bayesian optimization (BO) and evolutionary search (ES) are popular strategies for NAS (e.g., [21, 24, 43, 50, 37]; [28, 12, 3, 17]). We are not aware of any work that successfully applies BO to one-shot NAS for tabular data; Guo et al. [18] proposed an ES approach for one-shot NAS on vision datasets. BO and ES omit the extra forward passes for RL controller training, but need extra forward passes to evaluate child networks from SuperNet weights (Figure 4 in the main paper). Thus we control the number of forward passes for a fair comparison[12]. We design the following one-shot NAS methods to compare with an $M$-epoch TabNAS that interleaves weight and RL updates in its latter 75% iterations:

- **train-then-search-with-RL**: Train the SuperNet for $M$ epochs as in TabNAS, then do NAS for $0.75M$ epochs with rejection-based RL (with each iteration as Algorithm 2).
- **train-then-search-with-BO**: Train the SuperNet for $M$ epochs as in TabNAS, then do BO (by Gaussian processes [35] with expected improvement [32, 22]) in the set of feasible architectures with a similar number of SuperNet forward passes for child network evaluation.
- **train-then-search-with-ES**: Train the SuperNet for $M$ epochs as in TabNAS, then do ES in the set of feasible architectures as Algorithm 1 in Guo et al. [18] with a similar number of SuperNet forward passes for child network evaluation: in each iteration, start with population size $P$, pick top-$k$ architectures, crossover and mutate these top-$k$ to each get $P/2$ architectures, then combine the crossover and mutation results to get the population for the next iteration.

On Criteo, the cost of forward passes for RL is comparable to evaluating 405 child networks on the validation set. The search space of 5-layer FFNs has 340,590 feasible architectures below the 75,353 parameters limit (corresponding reference architecture is 48-240-24-256-8) in Figure A2(b). In BO, we tune RBF kernel length scale, number of initially sampled architectures, and number of new architectures to sample in each step; in ES, we tune population size and $k$. We can see from the results in Table A5 that:

---

[12]RL also does backward passes to optimize over the logits $\{\ell_{ij}\}_{i\in[L], j\in[C_i]}$, but the number of logits in our setting is much fewer than the size of the validation set (100 vs. 4,582,432), which means the cost of forward passes dominates.

Table A4: Architectures found by RL with 5 reward functions on 5 tabular datasets in Table A1. Each reward-dataset combination has 3 repeated NAS runs. The bracket "(2 times)" indicates the same architecture was found by the corresponding reward function twice.

| | Criteo | Volkert | Aloi | Connect-4 | Higgs |
|---|---|---|---|---|---|
| reference architecture | **32-144-24 (41,153 parameters, 0.4454 ± 0.0003)** | **48-160-32-144 (27,882 parameters, 0.3244 ± 0.0040)** | 160-64-512-64 (162,056 parameters, 0.0470 ± 0.0004) | **32-22-4-20 (3,701 parameters, 0.2691 ± 0.0021)** | **16-144-16 (5,265 parameters, 0.2794 ± 0.0013)** |
| RL with rejection-based reward | **32-144-24 (2 times) (41,153 parameters, 0.4454 ± 0.0003) 32-112-32 (40,241 parameters, 0.4456 ± 0.0003)** | **64-128-48-16 (27,050 parameters, 0.3237 ± 0.0040) 48-160-32-144 (27,882 parameters, 0.3244 ± 0.0040) 80-48-112-32 (27,802 parameters, 0.3274 ± 0.0037)** | **176-144-144-80 (2 times) (161,672 parameters, 0.0458 ± 0.0007) 192-112-176-80 (161,432 parameters, 0.0461 ± 0.0012)** | 20-32-14-48 (3,701 parameters, 0.2827 ± 0.0062) 20-30-24-22 (3,693 parameters, 0.2795 ± 0.0049) 28-10-12-56 (3,701 parameters, 0.2819 ± 0.0052) | 72-24-48 (2 times) (5,161 parameters, 0.2911 ± 0.0031) 80-32-8 (5,265 parameters, 0.2893 ± 0.0022) 64-16-128 (5,265 parameters, 0.2870 ± 0.0033) |
| RL with Abs Reward | 32-64-96 (2 times) (41,345 parameters, 0.4461 ± 0.0003) 32-80-64 (40,785 parameters, 0.4459 ± 0.0002) | 96-64-32-48 (27,738 parameters, 0.3302 ± 0.0042) 96-48-32-96 (27,738 parameters, 0.3305 ± 0.0047) 96-80-16-48 (27,738 parameters, 0.3302 ± 0.0050) | 144-112-144-96 (3 times) (162,008 parameters, 0.0473 ± 0.0004) 128-96-96-112 (4 times) (162,072 parameters, 0.0488 ± 0.0012) 112-112-96-112 (161,816 parameters, 0.0502 ± 0.0007) | **30-22-12-12 (3,703 parameters, 0.2702 ± 0.0027)** 26-20-18-26 (2 times) (3,703 parameters, 0.2807 ± 0.0029) 26-18-20-26 (3,703 parameters, 0.2765 ± 0.0046) | 104-16-24 (2 times) (5,233 parameters, 0.2896 ± 0.0036) 80-16-88 (2 times) (5,281 parameters, 0.2887 ± 0.0021) 88-16-64 (2 times) (5,217 parameters, 0.2874 ± 0.0027) |
| MNasNet (reward type 1: $Q(y)(T(y)/T_0)^\beta$) | 32-64-16 (36,065 parameters, 0.4464 ± 0.0003) 32-64-8 (35,537 parameters, 0.4466 ± 0.0002) 32-24-64 (35,353 parameters, 0.4466 ± 0.0003) | only found one feasible architecture: 32-32-224-24 (19,890 parameters, 0.3392 ± 0.0043) | 160-128-112-96 (163,544 parameters, 0.0469 ± 0.0009) 128-96-96-112 (162,072 parameters, 0.0488 ± 0.0012) 128-176-128-80 (153,192 parameters, 0.0473 ± 0.0007) | 24-18-16-44 (3,677 parameters, 0.2788 ± 0.0046) 28-12-32-14 (3,651 parameters, 0.2781 ± 0.0052) 22-32-16-22 (3,577 parameters, 0.2818 ± 0.0032) | 72-24-32 (4,745 parameters, 0.2898 ± 0.0019) 64-32-16 (4,545 parameters, 0.2888 ± 0.0018) 72-24-24 (2 times) (4,537 parameters, 0.2903 ± 0.0036) |
| MNasNet (reward type 2: $Q(y)\max\{1,(T(y)/T_0)^\beta\}$) | 24-128-16 (29,953 parameters, 0.4468 ± 0.0005) 16-176-48 (27,985 parameters, 0.4478 ± 0.0006) 24-16-160 (27,953 parameters, 0.4479 ± 0.0003) | only found one feasible architecture: 80-24-24-24 (17,874 parameters, 0.3521 ± 0.0013) | 112-144-240-64 (145,944 parameters, 0.0480 ± 0.0005) 160-112-128-64 (126,392 parameters, 0.0497 ± 0.0019) 256-64-224-32 (104,232 parameters, 0.0548 ± 0.0015) | 20-12-44-24 (3,679 parameters, 0.2871 ± 0.0069) 28-16-10-52 (3,745 parameters, 0.2776 ± 0.0061) 14-40-28-20 (3,581 parameters, 0.2886 ± 0.0047) | 16-8-16 (2 times) (777 parameters, 0.2929 ± 0.0020) 8-8-24 (553 parameters, 0.3003 ± 0.0031) |
| RENA [51] | 32-16-48 (34,289 parameters, 0.4472 ± 0.0002) 24-48-160 (33,873 parameters, 0.4468 ± 0.0001) 8-64-96 (15,137 parameters, 0.4523 ± 0.0004) | only found one feasible architecture: 32-48-24-512 (26,482 parameters, 0.3389 ± 0.0039) | didn't find any (close to) feasible architecture | 24-26-20-16 (3,617 parameters, 0.2806 ± 0.0068) 30-18-10-32 (3,749 parameters, 0.2752 ± 0.0075) 26-26-18-8 (3,577 parameters, 0.2769 ± 0.0031) | only found one feasible architecture: 80-8-152 (4,569 parameters, 0.2882 ± 0.0016) |

Table A5: TabNAS vs. train-then-search-with-RL/BO/ES for one-shot NAS on Criteo, among 5-layer FFNs. The reference architecture 48-240-24-256-8 has 75,353 parameters and validation loss 0.4448 $\pm$ 0.0002. Bold results below indicate architectures that are on par with the best validation loss 0.4444 $\pm$ 0.0002. We run each method 3 times, except for TabNAS, which was detailed in Appendix D. The square bracket "[3 times]" indicates that the same architecture was found by the corresponding method three times.

| method | found architectures (number of parameters, mean $\pm$ std loss) | NAS cost |
|---|---|---|
| TabNAS ($\eta = 0.005$ or $0.001$, $N = 32768$) | **48-176-64-16-256 (74,945 parameters, 0.4445 $\pm$ 0.0002)**
**48-208-48-48-64 (75,121 parameters, 0.4444 $\pm$ 0.0001)**
**48-176-80-16-96 (75,153 parameters, 0.4445 $\pm$ 0.0002)** | $1.86 \times 10^9$ forward passes (405 child network evaluations) |
| train-then-search-with-RL ($\eta = 0.0005$ or $0.001$, $N = 32768$) | **48-64-224-16-256 (75,249 parameters, 0.4446 $\pm$ 0.0002)**
**48-240-16-24-384 (75,353 parameters, 0.4445 $\pm$ 0.0002)**
**48-176-64-16-256 (74,945 parameters, 0.4445 $\pm$ 0.0002)** | $1.86 \times 10^9$ forward passes (405 child network evaluations) |
| train-then-search-with-BO (RBF kernel L=10, white noise variance 1, 50 initial samples, 20 new samples per BO iteration) | 64-80-8-16-16 (72,073 parameters, 0.4447 $\pm$ 0.0002)
**48-384-16-16-16 (74,881 parameters, 0.4444 $\pm$ 0.0002)**
48-256-16-8-240 (68,537 parameters, 0.4447 $\pm$ 0.0002) | 153 child network evaluations, then gets stuck |
| train-then-search-with-BO (RBF kernel L=1, white noise variance 1, 100 initial samples, 10 new samples per BO iteration) | 48-80-48-96-96 (71,265 parameters, 0.4451 $\pm$ 0.0003)
48-128-8-64-8 (57,753 parameters, 0.4451 $\pm$ 0.0002)
**64-96-16-32-16 (74,673 parameters, 0.4446 $\pm$ 0.0002)** | 420 child network evaluations |
| train-then-search-with-BO (RBF kernel L=1, white noise variance 1, 50 initial samples, 20 new samples per BO iteration) | **64-112-16-8-16 (75,177 parameters, 0.4445 $\pm$ 0.0001)**
48-32-256-8-240 (63,817 parameters, 0.4454 $\pm$ 0.0003)
64-96-16-24-48 (75,241 parameters, 0.4447 $\pm$ 0.0002) | 430 child network evaluations |
| train-then-search-with-BO (RBF kernel L=0.1, white noise variance 1, 50 initial samples, 20 new samples per BO iteration) | [3 times] 64-96-16-24-48 (75,241 parameters, 0.4447 $\pm$ 0.0002) | 430 child network evaluations |
| train-then-search-with-BO (RBF kernel L=0.1, white noise variance 1, 100 initial samples, 10 new samples per BO iteration) | [3 times] 64-96-16-24-48 (75,241 parameters, 0.4447 $\pm$ 0.0002) | 430 child network evaluations |
| train-then-search-with-ES (population 50, $k = 10$) | 48-96-32-224-16 (68,161 parameters, 0.4450 $\pm$ 0.0003)
48-48-8-160-64 (63,897 parameters, 0.4455 $\pm$ 0.0003)
48-96-32-24-64 (59,609 parameters, 0.4448 $\pm$ 0.0001) | 424 child network evaluations |
| train-then-search-with-ES (population 25, $k = 20$) | 48-144-32-16-144 (64,161 parameters, 0.4447 $\pm$ 0.0002)
48-80-16-128-64 (65,057 parameters, 0.4453 $\pm$ 0.0002)
48-96-32-48-64 (61,937 parameters, 0.4451 $\pm$ 0.0001) | 469 child network evaluations |

- RL-based methods (TabNAS or train-then-search-with-RL) stably finds architectures that are qualitatively the best.

- There is a large variance across architectures found by each hyperparameter setting of BO or ES. The search results are sensitive to initialization and are worse than those found by RL in over $2/3$ trials. We observe that each of the BO and ES searches quickly gets stuck at an architecture that is close to the best architecture in the initial set of samples.

- The local optima that BO and ES get stuck at still often have bottleneck structures, but the number of parameters is often significantly below the limit (e.g., 63,817 parameters in the searched model vs. a limit of 75,353 parameters). Model performance suffers as a result.

Many interesting questions on BO and ES for one-shot NAS remain open for future work, including how to control the initialization randomness (for better exploration of the search space) and how to design methods that properly interleave weight training and NAS steps under such large exploration randomness (for better exploitation of promising architectures).

Table A6: #FLOPs (M) of architectures found by RL with the rejection-based reward and the Abs Reward in NATS-Bench size search space.

| RL learning rate | 0.01 | 0.05 | 0.1 | 0.5 |
|---|---|---|---|---|
| rejection-based reward, $N$=200 | $74.7 \pm 0.3$ (median 74.7) | $74.7 \pm 0.2$ (median 74.7) | $74.6 \pm 0.3$ (median 74.7) | $70.7 \pm 4.5$ (median 72.4) |
| Abs Reward, $\beta$=-10 | $77.0 \pm 12.7$ (median 76.4) | $75.1 \pm 4.5$ (median 74.8) | $75.2 \pm 1.7$ (median 75.1) | $75.7 \pm 3.8$ (median 75.2) |
| Abs Reward, $\beta$=-5 | $78.1 \pm 13.1$ (median 77.0) | $75.6 \pm 4.3$ (median 75.5) | $75.4 \pm 1.7$ (median 75.2) | $76.0 \pm 4.4$ (median 75.3) |
| Abs Reward, $\beta$=-2 | $80.6 \pm 13.8$ (median 78.5) | $76.3 \pm 4.2$ (median 76.1) | $75.7 \pm 1.8$ (median 75.5) | $75.9 \pm 5.4$ (median 75.4) |
| Abs Reward, $\beta$=-1 | $83.0 \pm 13.3$ (median 81.9) | $77.4 \pm 4.4$ (median 77.1) | $76.0 \pm 2.1$ (median 75.8) | $76.7 \pm 5.3$ (median 75.6) |
| Abs Reward, $\beta$=-0.5 | $87.1 \pm 12.7$ (median 86.7) | $78.8 \pm 4.5$ (median 78.7) | $77.0 \pm 2.6$ (median 76.6) | $76.9 \pm 6.0$ (median 76.1) |
| Abs Reward, $\beta$=-0.2 | $95.5 \pm 12.8$ (median 95.3) | $80.6 \pm 5.2$ (median 80.1) | $78.5 \pm 3.6$ (median 77.9) | $78.5 \pm 9.2$ (median 76.8) |
| Abs Reward, $\beta$=-0.1 | $101.8 \pm 13.3$ (median 103.0) | $84.2 \pm 7.6$ (median 83.0) | $81.1 \pm 5.8$ (median 80.5) | $80.1 \pm 10.6$ (median 78.6) |

Table A7: Test errors of architectures found by RL with the rejection-based reward and the Abs Reward in NATS-Bench size search space.

| RL learning rate | 0.01 | 0.05 | 0.1 | 0.5 |
|---|---|---|---|---|
| rejection-based reward, $N$=200 | $0.468 \pm 0.050$ | $0.435 \pm 0.023$ | $0.425 \pm 0.014$ | $0.420 \pm 0.008$ |
| Abs Reward, $\beta$=-10 | $0.483 \pm 0.056$ | $0.468 \pm 0.034$ | $0.473 \pm 0.046$ | $0.490 \pm 0.074$ |
| Abs Reward, $\beta$=-5 | $0.474 \pm 0.049$ | $0.461 \pm 0.029$ | $0.463 \pm 0.039$ | $0.481 \pm 0.061$ |
| Abs Reward, $\beta$=-2 | $0.462 \pm 0.039$ | $0.450 \pm 0.022$ | $0.455 \pm 0.028$ | $0.468 \pm 0.046$ |
| Abs Reward, $\beta$=-1 | $0.449 \pm 0.027$ | $0.442 \pm 0.016$ | $0.445 \pm 0.020$ | $0.456 \pm 0.035$ |
| Abs Reward, $\beta$=-0.5 | $0.436 \pm 0.019$ | $0.434 \pm 0.014$ | $0.435 \pm 0.015$ | $0.444 \pm 0.023$ |
| Abs Reward, $\beta$=-0.2 | $0.422 \pm 0.013$ | $0.424 \pm 0.011$ | $0.426 \pm 0.012$ | $0.433 \pm 0.016$ |
| Abs Reward, $\beta$=-0.1 | $0.414 \pm 0.009$ | $0.416 \pm 0.008$ | $0.418 \pm 0.009$ | $0.426 \pm 0.014$ |

# G   Rejection-based reward outperforms Abs Reward in NATS-Bench size search space (full version)

In the vision domain, the NATS-Bench [10] size search space has convolutional network architectures with a predefined skeleton and 8 candidate sizes {8, 16, 24, 32, 40, 48, 56, 64} for each of its 5 layers. In this search space with $8^5 = 32768$ candidate architectures, we use the true number of floating point operations (#FLOPs) as our cost metric, validation accuracy on CIFAR-100 [27] as RL quality reward, and test error (1 - test accuracy) on CIFAR-100 as final performance metric (the use of error instead of accuracy is consistent with other results here). We use 75M #FLOPs as the resource limit, so that there are 13,546 (41.3%) feasible architectures in the search space. This experiment setting is the same as in the toy example (Figure 1), where we regard the network weights as given and only compare the RL controllers. We do grid search over NAS hyperparameters: {0.01, 0.05, 0.1, 0.5} for the RL learning rate for both the rejection-based reward and the Abs Reward, and {10, 5, 2, 1, 0.5, 0.2, 0.1} for $|\beta|$ in the Abs Reward. We show #FLOPs and test error statistics (mean $\pm$ std) of architectures found by the rejection-based RL reward and the Abs Reward across 500 NAS repetitions of each experiment setting.

Detailed results of #FLOPs and test errors of architectures found by either of the RL rewards at different NAS hyperparameters are listed in Table A6 and A7, respectively.

We can see that:

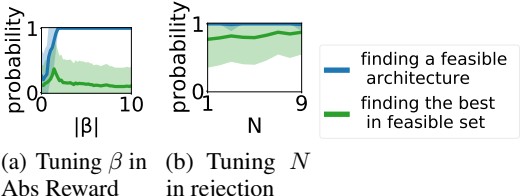

(a) Tuning $\beta$ in Abs Reward

(b) Tuning $N$ in rejection

Figure A4: **Tuning $\beta$ and $N$ on the toy example (Figure 1)**: the number of MC samples $N$ in rejection-based reward is easier to tune than $\beta$ in Abs Reward, and is easier to succeed. The lines and shaded regions are mean and standard deviation across 200 independent runs, respectively.

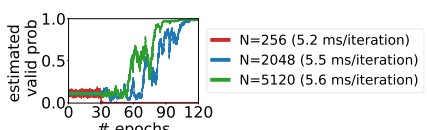

Figure A5: **Tuning $N$ on Criteo**: the change of $\widehat{\mathbb{P}}(V)$ when the number of Monte-Carlo samples $N$ is 256, 2,048 or 5,120, and the time taken for each iteration. We show results with RL learning rate $\eta = 0.005$; those other $\eta$ values have similar failure patterns.

- RL with the rejection-based reward finds architectures that are within the 75M #FLOPs limit.
- When $|\beta|$ is large, the architectures found by RL with the Abs Reward are within the 75M #FLOPs limit, but they are inferior in quality (have larger test errors).
- When $|\beta|$ is small, the architectures found by RL with the Abs Reward are similar or better in quality, but they exceed the 75M #FLOPs limit by >5%.

These observations are similar to what we saw in the toy example, showing the rejection-based reward outperforms in tasks from both tabular and vision domains.

Taking a closer look at the architectures that are Pareto-optimal and those found by each RL reward, we can see that:

- The Pareto-optimal architectures often have bottleneck structures like their counterparts in tabular tasks; examples include (in NATS-Bench format, and same below) 8:32:16:40:48, 8:56:40:56:64, and 16:56:64:64:56 that have a bottleneck in their first channel.
- Bottleneck structures occur less often in the architectures found by RL with the Abs Reward. Examples include 24:24:64:40:32, 24:40:56:40:48, and 32:48:40:32:32. The architectures found by RL with the rejection-based reward (our method) have more similar appearances as the Pareto-optimal architectures; examples include 8:56:64:64:64, 8:40:56:64:56, and 16:64:48:64:64.

This means the co-adaptation problem (described in Section 1 of the main paper) also occurs in RL-based NAS with resource-aware rewards in the image domain.

## H  Difficulty of hyperparameter tuning

Hyperparameter tuning has always been a headache for machine learning. In the design of NAS approaches, the hope is that the NAS hyperparameters are much easier to tune than the architectures NAS search over. We denote the RL learning rate and the number of MC samples by $\eta$ and $N$, respectively. The three resource-aware rewards (in MnasNet and TuNAS) have both $\eta$ and $\beta$ as hyperparameters; our TabNAS with the rejection-based reward has $\eta$ and $N$ to tune.

### H.1  Resource hyperparameter $\beta$

$\beta$ is difficult to tune in experiments: the best value varies by dataset and lies in the middle of its search space. Since $\beta < 0$, we discuss its absolute value. In a NAS search space, the architecture that is feasible and can match the reference performance often has the number of parameters that is more than 98% of the reference. A too small $|\beta|$ is not powerful enough to enforce the resource constraint, in which case NAS finds an architecture that is far from the target number of parameters and makes the search nearly unconstrained (e.g., the Abs Reward with $|\beta| = 1$ in the toy example, shown in Figure 1 and towards the left end in Figure A4(a)). A too large $|\beta|$ severely penalizes the violation of the resource constraint, in which case the RL controller would always give an architecture close to the reference, with much bias (e.g., the Abs Reward with $|\beta| = 2$ in Figure 1, and towards the right end in Figure A4(a)). Thus practitioners seek a medium $|\beta|$ in hyperparameter tuning to both obey the resource constraint and achieve a better result. In our experiments, such "appropriate" medium

values vary largely across datasets: 1 on Criteo with the 32-144-24 reference architecture (41,153 parameters), 2 on Volkert with the 48-160-32-144 reference architecture (27,882 parameters), and 25 on Aloi with the 64-192-48-32 reference architecture (64,568 parameters).

## H.2   RL learning rate $\eta$

The RL learning rate $\eta$ is easier to tune and more generalizable across datasets than $\beta$. With a large $\eta$, the RL controller quickly converges right after the first $25\%$ epochs of layer warmup; with a small $\eta$, the RL controller converges slowly or may not converge, although there may still be enough signal from the layerwise probabilities to get the final result. It is thus straightforward to tune $\eta$ by observing the convergence behavior of sampling probabilities. In our experiments, the appropriate value of $\eta$ does not significantly vary across tasks: a constant $\eta \in [0.001, 0.01]$ is appropriate for all datasets and all number of parameter limits.

## H.3   Number of MC samples $N$

The number of MC samples $N$ is also easier to tune than $\beta$. Resource permitting, $N$ is the larger, the better (Figure A4(b)), so that $\mathbb{P}(V)$ can be better estimated. When $N$ is too small, the MC sampling has a high chance of missing the valid architectures in the search space, and thus incurs large bias and variance for the estimate of $\nabla \log[\mathbb{P}(y \mid y \in V)]$. In such cases, $\widehat{\mathbb{P}}(V)$ may miss all valid architectures at the beginning of RL and quickly converge to 0. $\widehat{\mathbb{P}}(V)$ being equal or close to 0 is a bad case for our rejection-based algorithm: the single-step RL objective $J(y)$ that has a $-\log(\widehat{\mathbb{P}}(V))$ term grows extremely large and gives an explosive gradient to stuck the RL controller in the current choice. Consequently, the criterion for choosing $N$ is to choose the largest that can afford, and hopefully, at least choose the smallest that can make $\widehat{\mathbb{P}}(V)$ steadily increase during RL. Figure A5 shows the changes of $\widehat{\mathbb{P}}(V)$ on Criteo with the 32-144-24 reference in the search space of 8,000 architectures at three $N$ values. The NAS succeeds when $N \geq 2048$, same as the threshold that makes $\widehat{\mathbb{P}}(V)$ increase.

Overall, the RL controller with our rejection-based reward has hyperparameters that are easier to tune than with resource-aware rewards in MnasNet and TuNAS.

# I   Ablation studies

We do the ablation studies on Criteo with the 32-144-24 reference. The behavior on other datasets with other reference architectures are similar.

**Whether to use $\widehat{\mathbb{P}}(V)$ instead of $\mathbb{P}(V)$.**   The Monte-Carlo (MC) sampling estimates $\mathbb{P}(V)$ with $\widehat{\mathbb{P}}(V)$ to save resources. Such estimations are especially efficient when the sample space is large. Empirically, the $\widehat{\mathbb{P}}(V)$ estimated with enough MC samples (as described in Appendix H) enables the RL controller to find the same architecture as $\mathbb{P}(V)$, because the $\widehat{\mathbb{P}}(V)$ estimated with a large enough number of samples is accurate enough (e.g., Figure 5(b) and 9(d)).

**Whether to skip infeasible architectures in weight updates.**   In each iteration of one-shot training and REINFORCE (Appendix B Algorithm 1) with the rejection mechanism (Appendix B Algorithm 2), we train the weights in the sampled child network $x$ regardless of whether $x$ is feasible. Instead, we may update the weights only when $x$ is feasible, in a similar rejection mechanism as the RL step. We find this mechanism may mislead the search because of insufficiently trained weights: the rejection-based RL controller can still find qualitatively the best architectures on Criteo with the 32-144-24 or 48-240-24-256-8 reference, but fails with the 48-128-16-112 reference. In the latter case, although the RL controller still finds architectures with bottleneck structures (e.g., 32-384-8-144), the first layer sizes of the found architectures are much smaller, leading to suboptimal performance.

**Whether to differentiate through $\widehat{\mathbb{P}}(V)$.**   Recall that REINFORCE with rejection has the objective

$$J(y) = \text{stop\_grad}(Q(y) - \overline{Q}) \cdot \log\left[\mathbb{P}(y)/\mathbb{P}(V)\right].$$

To update the RL controller's logits, we compute $\nabla J(y)$, which requires a differentiable approximation of $\mathbb{P}(V)$. From a theoretical standpoint, omitting the extra term $\mathbb{P}(V)$ – or using a non-

differentiable approximation – will result in biased gradient estimates. Empirically, we ran experiments with multiple variants of our algorithm where we omitted the term $\mathbb{P}(V)$, but found that the quality of the searched architectures was significantly worse.

Below are our experimental findings:

- In the case that we do not skip infeasible architectures in weight updates, the largest hidden layer sizes may gain and maintain the largest sampling probabilities soon after RL starts. This is because most architectures in the 3-layer Criteo search space are above the number of parameters limit 41,153. When RL starts, the sampled feasible architectures underperform the moving average, thus their logits are severely penalized, making the logits of the infeasible architectures (which often have wide hidden layers) quickly dominate (Figure A6(a)). Accordingly, the (estimated) valid probability $\mathbb{P}(V)$ (or $\widehat{\mathbb{P}}(V)$) quickly decrease to 0 (Figure A6(b)), and the RL controller gets stuck (as described in Appendix H.3) in these large choices for hidden layer sizes.

- In the case that we skip infeasible architectures in both weight and RL updates, the RL controller eventually picks feasible architectures with bottleneck structures, but the found architectures are almost always suboptimal: when RL starts, the controller severely boosts the logits of the sampled feasible architectures without much exploration in the search space, and quickly gets stuck there. For example, the search in Figure A6(c)) finds 24-384-16 (40,449 parameters) that is feasible but suboptimal; $\mathbb{P}(V)$ and $\widehat{\mathbb{P}}(V)$ quickly increase to 1 after RL starts (Figure A6(d)).

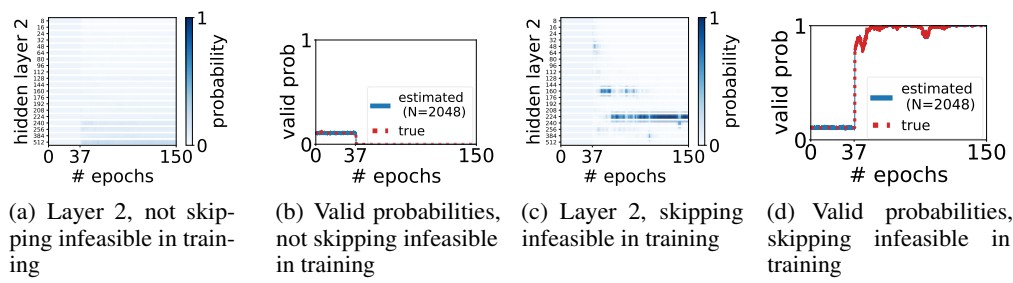

(a) Layer 2, not skipping infeasible in training

(b) Valid probabilities, not skipping infeasible in training

(c) Layer 2, skipping infeasible in training

(d) Valid probabilities, skipping infeasible in training

Figure A6: **Failure cases in ablation when $\widehat{\mathbb{P}}(V)$ is non-differentiable**. We show results with RL learning rate $\eta = 0.005$; those under other $\eta$ values are similar.

**Strategy for choosing the final architecture after search.** When RL finishes, instead of biasing towards architectures with more parameters (Appendix B Algorithm 3), we may also bias towards those that are feasible and have larger sampling probabilities. We find that when the final distributions are less deterministic, the architectures found by the latter strategy to perform worse: for example, the top 3 feasible architectures found with the final distribution in Figure 9 are 32-128-16, 32-160-16 and 32-128-8, and they are all inferior to 32-144-24.

# J Proofs

## J.1 $\widehat{\mathbb{P}}(V)$ is an unbiased and consistent estimate of $\mathbb{P}(V)$

Within the search space $S$, recall the definitions of $\mathbb{P}(V)$ and $\widehat{\mathbb{P}}(V)$:

- $\mathbb{P}(V) = \sum\limits_{z^{(i)} \in S} p^{(i)} \mathbb{1}(z^{(i)} \in V)$

- $\widehat{\mathbb{P}}(V) = \frac{1}{N} \sum\limits_{k \in [N], z^{(k)} \in V} \frac{p^{(k)}}{q^{(k)}} = \frac{1}{N} \sum\limits_{k \in [N]} \frac{p^{(k)}}{q^{(k)}} \mathbb{1}(z^{(k)} \in V)$

**Unbiasedness.** With $N$ architectures sampled from the proposal distribution $q$, we take the expectation with respect to $N$ sampled architectures:

$$\mathbb{E}[\widehat{\mathbb{P}}(V)] = \frac{1}{N}\mathbb{E}\left[\sum_{k\in[N],z^{(k)}\in V}\frac{p^{(k)}}{q^{(k)}}\right]$$

$$= \frac{1}{N}\mathbb{E}\left[\sum_{k\in[N]}\frac{p^{(k)}}{q^{(k)}}\mathbb{1}(z^{(k)}\in V)\right]$$

$$= \frac{1}{N}\sum_{k\in[N]}\mathbb{E}\left[\frac{p^{(k)}}{q^{(k)}}\mathbb{1}(z^{(k)}\in V)\right],$$

in which each summand

$$\mathbb{E}\left[\frac{p^{(k)}}{q^{(k)}}\mathbb{1}(z^{(k)}\in V)\right] = \sum_{z^{(k)}\in S}q^{(k)}\frac{p^{(k)}}{q^{(k)}}\mathbb{1}(z^{(k)}\in V)$$

$$= \mathbb{P}(V),$$

Thus $\mathbb{E}[\widehat{\mathbb{P}}(V)] = \mathbb{P}(V)$.

**Consistency.** We first show the variance of $\mathbb{P}(V)$ converges to 0 as the number of MC samples $N \to \infty$. Because of independence among samples,

$$\mathrm{Var}[\widehat{\mathbb{P}}(V)] = \frac{1}{N}\sum_{k\in[N]}\mathrm{Var}\left[\frac{p^{(k)}}{q^{(k)}}\mathbb{1}(z^{(k)}\in V)\right],$$

in which each summand

$$\mathrm{Var}\left[\frac{p^{(k)}}{q^{(k)}}\mathbb{1}(z^{(k)}\in V)\right] = \mathbb{E}\left[\frac{p^{(k)}}{q^{(k)}}\mathbb{1}(z^{(k)}\in V) - \mathbb{P}(V)\right]$$

$$= \sum_{z^{(k)}\notin V}q^{(k)}\mathbb{P}(V)^2 + \sum_{z^{(k)}\in V}q^{(k)}\left[\frac{p^{(k)}}{q^{(k)}} - \mathbb{P}(V)\right]^2 \qquad \text{(A1)}$$

$$= -\mathbb{P}(V)^2 + \sum_{z^{(k)}\in V}\frac{(p^{(k)})^2}{q^{(k)}},$$

thus the variance

$$\mathrm{Var}[\widehat{\mathbb{P}}(V)] = \frac{1}{N}\sum_{k\in[N]}\mathrm{Var}\left[\frac{p^{(k)}}{q^{(k)}}\mathbb{1}(z^{(k)}\in V)\right]$$

$$\frac{1}{N}\left[-\mathbb{P}(V)^2 + \sum_{z^{(k)}\in V}\frac{(p^{(k)})^2}{q^{(k)}}\right],$$

which goes to 0 as $N \to \infty$. It worths noting that when we set $q = \mathrm{stop\_grad}(p)$, the single-summand variance (Equation A1) becomes $\mathbb{P}(V) - \mathbb{P}(V)^2$, which is the variance of a Bernoulli distribution with mean $\mathbb{P}(V)$.

The Chebyshev's Inequality states that for a random variable $X$ with expectation $\mu$, for any $a > 0$, $\mathbb{P}(|X - \mu| > a) \leq \frac{\mathrm{Var}(X)}{a^2}$. Thus $\lim_{N\to\infty}\mathrm{Var}(X) = 0$ implies that $\lim_{N\to\infty}\mathbb{P}(|X - \mu| > a) = 0$ for any $a > 0$, indicating consistency.

**J.2** $\nabla\log[\mathbb{P}(y)/\widehat{\mathbb{P}}(V)]$ **is a consistent estimate of** $\nabla\log[\mathbb{P}(y\,|\,y\in V)]$

Since $\mathbb{P}(y\,|\,y\in V) = \frac{\mathbb{P}(y)}{\mathbb{P}(V)}$, we show $\mathrm{plim}_{N\to\infty}\nabla\log\widehat{\mathbb{P}}(V) = \nabla\log\mathbb{P}(V)$ below to prove consistency, in which $\mathrm{plim}_{N\to\infty}$ denotes convergence in probability.

Recall $p^{(i)}$ is the probability of sampling the $i$-th architecture $z^{(i)}$ within the search space $S$, and the definitions of $\mathbb{P}(V)$ and $\widehat{\mathbb{P}}(V)$ are:

- $\mathbb{P}(V) = \sum\limits_{z^{(i)} \in S} p^{(i)} \mathbb{1}(z^{(i)} \in V),$

- $\widehat{\mathbb{P}}(V) = \frac{1}{N} \sum\limits_{k \in [N], z^{(k)} \in V} \frac{p^{(k)}}{q^{(k)}} = \frac{1}{N} \sum\limits_{k \in [N]} \frac{p^{(k)}}{q^{(k)}} \mathbb{1}(z^{(k)} \in V),$ in which each $p^{(k)}$ is differentiable with respect to all logits $\{\ell_{ij}\}_{i \in [L], j \in [C_i]}$.

Thus we have

$$\mathop{\mathrm{plim}}\limits_{N \to \infty} \widehat{\mathbb{P}}(V) = \mathop{\mathrm{plim}}\limits_{N \to \infty} \frac{1}{N} \sum\limits_{k \in [N]} \frac{p^{(k)}}{q^{(k)}} \mathbb{1}(z^{(k)} \in V)$$

$$= \frac{1}{N} \sum\limits_{z^{(k)} \in S} \frac{p^{(k)}}{q^{(k)}} \mathbb{1}(z^{(k)} \in V) N q^{(k)}$$

$$= \sum\limits_{z^{(k)} \in S} p^{(k)} \mathbb{1}(z^{(k)} \in V) = \mathbb{P}(V),$$

and

$$\mathop{\mathrm{plim}}\limits_{N \to \infty} \nabla\widehat{\mathbb{P}}(V) = \mathop{\mathrm{plim}}\limits_{N \to \infty} \frac{1}{N} \sum\limits_{k \in [N]} \frac{\nabla p^{(k)}}{q^{(k)}} \mathbb{1}(z^{(k)} \in V)$$

$$= \frac{1}{N} \sum\limits_{z^{(k)} \in S} \frac{\nabla p^{(k)}}{q^{(k)}} \mathbb{1}(z^{(k)} \in V) N q^{(k)}$$

$$= \sum\limits_{z^{(k)} \in S} \nabla p^{(k)} \mathbb{1}(z^{(k)} \in V) = \nabla\mathbb{P}(V).$$

Together with the condition that $\mathbb{P}(V) > 0$ (the search space contains at least one feasible architecture), we have the desired result for consistency as $\mathop{\mathrm{plim}}\limits_{N \to \infty} \nabla \log \widehat{\mathbb{P}}(V) = \mathop{\mathrm{plim}}\limits_{N \to \infty} \frac{\nabla\widehat{\mathbb{P}}(V)}{\widehat{\mathbb{P}}(V)} = \frac{\mathop{\mathrm{plim}}\limits_{N \to \infty} \nabla\widehat{\mathbb{P}}(V)}{\mathop{\mathrm{plim}}\limits_{N \to \infty} \widehat{\mathbb{P}}(V)} = \frac{\nabla\mathbb{P}(V)}{\mathbb{P}(V)} = \nabla \log \mathbb{P}(V),$ in which the equalities hold due to the properties of convergence in probability.