# OpenReview forum: "TabNAS: Rejection Sampling for Neural Architecture Search on Tabular Datasets"
_NeurIPS.cc/2022/Conference — NeurIPS 2022 Accept_

### Official Review · Reviewer_w3th · 2022-07-03

**Rating:** 7
**Confidence:** 2
**Soundness:** 4 excellent
**Presentation:** 4 excellent
**Contribution:** 4 excellent

**Summary:**

This paper proposes a neural architecture search algorithm designed for tabular datasets. Prior work uses an RL-based optimization procedure with penalties for feasibility and resource constraints, but the authors show that this does not work well. To address this issue, the authors add an additional rejection sampling phase to filter out those that do not satisfy constraints, and show that this allows the RL method to better explore the search space of architectures. Evaluation is done on the Criteo and Volkert datasets, where the found architectures consistently achieve lower losses.

**Questions:**

I think the results could be summarized in a more concise and clear way (e.g. a table) - in the paper the only results are shown in 2 small graphs where it's difficult to tell what the absolute numbers are and which baseline is being run.

**Limitations:**

The authors did not discuss any potential negative societal impacts of their work.

**Strengths And Weaknesses:**

Strengths:
- I believe that the use of rejection sampling in NAS algorithms is novel. However, I am not familiar with neural architecture search so my knowledge of related work is relatively small.
- The work is significant, in that it unlocks more NAS applications in a new domain area.
- In general, the writing is clear and well-organized - the authors clearly present their method and reasoning, and I appreciate that the authors experimented with multiple baselines and multiple datasets and found consistent results between them.

Weaknesses:
- The experimental results are presented on a domain the baseline methods were not developed for, so it is more difficult to put the results in context. It would be informative to see how well the proposed methods work well on image-based domains, even if it does not perform as well.

---

> ### Author Response · Authors · 2022-08-02
> **Response to Reviewer w3th**
>
> Thank you for your careful reading and thoughtful reviews. We appreciate your recognition that “the use of rejection sampling in NAS algorithms is novel”, “(the work) unlocks more NAS applications in a new domain area”, and “the writing is clear and well-organized”.
>
> And we appreciate your suggestion that “it would be informative to see how well the proposed methods work well on image-based domains”. Please see Response to All Reviewers for: (a) a comparison in the NATS-Bench size search space ([Part 1/3](https://openreview.net/forum?id=hHrO6-IfskR&noteId=15C32DfYrDA) and [Part 2/3](https://openreview.net/forum?id=hHrO6-IfskR&noteId=KwPTU0xzOrp)), and (b) a table that shows performance of different RL rewards across datasets ([Part 3/3](https://openreview.net/forum?id=hHrO6-IfskR&noteId=tJetVtOr81)). We will add these results to the revised version.

---

> > ### Comment · Reviewer_w3th · 2022-08-07
> > **Thank you for the response.**
> >
> > Thank you for the response. I appreciate the additional experimental results - the method appears to be strong even in the same domain used in the original Mnas/TuNas papers. I will maintain my rating of accept.

---

### Official Review · Reviewer_guLP · 2022-07-12

**Rating:** 6
**Confidence:** 4
**Soundness:** 3 good
**Presentation:** 3 good
**Contribution:** 3 good

**Summary:**

This paper proposes a multi-objective NAS algorithm using RL-controller. They use monte-carlo based rejection sampling to guide the controller towards feasible search space.
The search space is divided into possible categorical options for each layer, such as [16, 22, 42] etc for layer1. The controller predicts logits corresponding to each possible value for each layer in the fully connected network (FCNN). So logits_{ij} refers to the value for $j^{th}$ value in the $i^{th}$ layer and corresponds to the probability that it would be sampled. As the choice of each layer is independent of the other, to generate a network adhering to the constraints, they use rejection based reward mechanism.

**Questions:**

1) (a) Please tabulate the accuracies, number of parameters and time taken of networks obtained from MnasNet, TuNAS, TabNAS and for all the datasets.

(b) MoNas [1] is also an RL based multi-objective algorithm which yields a reward only if the constraint < threshold.  Resource-Efficient Neural Architect[2] uses a different penalty as listed in eqn 3 of their paper. Please compare how these two objectives fare against yours.

(c) While you are focusing only on RL based multi-objective algorithms, it would be good to compare with evolutionary based algos, namely Lemonade and NSGA-NET[3] .

 (d) It would also be good to compare it against TabNet. Based on the number of parameters of the best model found by Tabnet, you can modify your search to fit that parameter search space.

2) While rejection sampling is helping us focus the search on only the feasible area, the number of samples required to get a good estimate of the probability distribution seem to be very high. So it might not be scalable to larger search spaces. For ex,  5 * $10^{6}$ samples for 9-layer search space for Volkert dataset is computationally expensive and cannot be used in real world.


[1] MONAS: Multi-Objective Neural Architecture Search, Hsu et al.
[2] Resource-Efficient Neural Architect, Zhou et al.
[3] NSGA-Net: Neural Architecture Search using Multi-Objective Genetic Algorithm, Lu et al.

**Limitations:**

The monte carlo sampling is very expensive and this search won't scale for tasks requiring large search spaces. Please see Questions section for further details.

**Strengths And Weaknesses:**

They showed that their technique is able to find well performing networks.

However, they did not compare adequately against all the baselines. Tabulating the results rather than presenting them in a paragraph is preferred. Please see Questions for more details.

---

> ### Author Response · Authors · 2022-08-02
> **Response to Reviewer guLP**
>
> Thank you for your careful reading and thoughtful reviews. Let us address your comments below.
>
> > Please tabulate the accuracies, number of parameters and time taken of networks obtained from MnasNet, TuNAS, TabNAS and for all the datasets.
>
> Thanks for the suggestion. Please see [Response to All Reviewers - Part 3/3](https://openreview.net/forum?id=hHrO6-IfskR&noteId=tJetVtOr81) for the table. We will add the table to the revised version.
>
> > MoNas [1] is also an RL based multi-objective algorithm which yields a reward only if the constraint < threshold.
>
> Our RL reward can be understood as this hard constraint + probability reweighting, and we’ve compared it with only having a hard constraint in the ablation studies. Please see [Response to All Reviewers - Part 2/3](https://openreview.net/forum?id=hHrO6-IfskR&noteId=KwPTU0xzOrp).
>
> > Resource-Efficient Neural Architect[2] uses a different penalty as listed in eqn 3 of their paper.
>
> Please see the table in [Response to All Reviewers - Part 3/3](https://openreview.net/forum?id=hHrO6-IfskR&noteId=tJetVtOr81) for the comparison with the RL reward in RENA.
>
> > While you are focusing only on RL based multi-objective algorithms, it would be good to compare with evolutionary based algos, namely Lemonade and NSGA-NET
>
> Please see Appendix E for the comparison with evolutionary search in one-shot NAS. The multi-objective methods without weight sharing are significantly more expensive than the techniques we compare to, since they require full or partial training for multiple candidate architectures.
>
> > It would also be good to compare it against TabNet.
>
> We agree, and will cite and try to include a comparison with them in the revised version.
>
> > While rejection sampling is helping us focus the search on only the feasible area, the number of samples required to get a good estimate of the probability distribution seem to be very high … computationally expensive and cannot be used in real world.
>
> Our Monte-Carlo (MC) sampling subroutine may be less expensive than what people think of: for each sampled architecture, we only need to compute its cost metric (number of parameters in paper). This is much cheaper than network forward passes on the dataset. Also, we only need MC sampling after warmup and before the RL controller converges, which only accounts for <30% epochs in our experiments (Figure 9). Thus our search cost is only 40% more than RL with the Abs Reward (Figure A3).

---

> > ### Comment · Reviewer_guLP · 2022-08-10
> > **Thank you for your response**
> >
> > Your method is performing better than the baselines and the rejection sampling is a neat idea. I would like to increase my score.

---

### Official Review · Reviewer_7VoP · 2022-07-22

**Rating:** 5
**Confidence:** 5
**Soundness:** 2 fair
**Presentation:** 3 good
**Contribution:** 2 fair

**Summary:**

TabNAS proposes a modification to Reinforce which instead of modifying the reward function to incorporate resource constraints, estimates the feasible set of architectures (e.g. architectures less than some threshold/reference) and updates the probabilities of the controller accordingly by redistributing the probability mass over the feasible set as more trial-and-error progresses.

**Questions:**

1. I am quite confused by the contribution of this paper. The way I read it, the authors show that reward function modification as has been commonly done before is not as good as rejection sampling as proposed in this work in coming up with architectures that better meet resource constraints. I buy this and note that most Pareto-frontier aware algorithms which output the *entire* Pareto-frontier employ rejection sampling to constrain search to only those parts of the search space that the user wants (e.g. give me the Pareto-frontier estimate less than < 100 milliseconds inference latency, or between 200-300 milliseconds inference latency) See https://arxiv.org/abs/2105.01015. But there seems to be nothing specific in the method to tabular datasets. I can see applying this method as-is to any task (e.g. vision amd NLP). Is my understanding correct? If so, then it perhaps happened that the authors ran on tabular datasets to get the ball rolling? Note that one can use any of the excellent NAS benchmarks without using GPUs or large compute to validate this method. My suggestion will be NAS-Bench-Suite which has several different benchmarks under a single interface. https://arxiv.org/abs/2201.13396.

2. My second concern/comment is perhaps a bit unfair to this work: It is a bit baffling to me that a stateful RL algorithm like Reinforce has been widely applied (and continues to be used in the broader AutoML community) to a problem setting where one doesn't need to do long horizon credit assignment. NAS is inherently stateless, one immediately knows the result of their action (sampled architecture or in HPO the sampled hyperparameter). So 1-step RL like contextual bandits or Bayesian optimization will be just as efficient and probably more. Can such rejection constraints be added to BOHB, HyperBand, BANANAS, NASBO etc? If so how will they be different from those in https://arxiv.org/abs/2105.01015?

 3. Related to point 2 above: in the appendix lines 251-254, states that there are open questions on BO and ES for one-shot NAS. But is that necessary or one could train the supernet using the style of training in Once-For-All https://arxiv.org/abs/1908.09791 or use all the insights for training supernets as detailed in https://openreview.net/forum?id=Esd7tGH3Spl and then doing search via any search technique? Note OFA uses evolutionary search, the earlier ENAS paper used Reinforce https://arxiv.org/abs/1802.03268 and indeed the actual search technique varies across papers. In this respect it appears to me that the aim of the experiments in the appendix comparing across RL, BO and ES since BO and ES are being run on the feasible set only. So, a comparison (on tabular task) seems orthogonal to the aim of the paper which is to show a modification of specifically Reinforce to take into account resource constraints better than others who modify the reward function?


**Limitations:**

Yes.

**Strengths And Weaknesses:**

Strengths: Paper is clear and well-written. Literature has been adequately cited and discussed.

Weaknesses: Please see my comments in the questions section.

---

> ### Author Response · Authors · 2022-08-02
> **Response to Reviewer 7VoP - Part 1/2**
>
> Thank you for your careful reading and thoughtful reviews. Let us address your comments below.
>
> > I buy this and note that most Pareto-frontier aware algorithms which … or between 200-300 milliseconds inference latency)
>
> - Focusing on the entire Pareto frontier in other methods vs. focusing on the Pareto frontier near the resource constraint in our method: For some applications, we don't know in advance what our latency constraints should look like, so a technique which explores a large portion of the Pareto frontier is potentially quite valuable. But in other cases, product constraints dictate a precise latency/throughput target that we need to hit, and for those use-cases, latency-targeted searches can be a better fit. Our paper focuses on the latter.
> - Comparison with rejection sampling that only keeps those parts of the search space that the user wants: Please see [Response to All Reviewers - Part 2/3](https://openreview.net/forum?id=hHrO6-IfskR&noteId=KwPTU0xzOrp).
>
> > Note that one can use any of the excellent NAS benchmarks without using GPUs or large compute to validate this method. My suggestion will be …
>
> Thanks for the insightful comment. We have provided results in the size search space of NATS-Bench (part of NAS-Bench-Suite that includes a size search space, on which the co-adaptation problem we observed in Figure 1 and Section 4.1 occurs more naturally than on operation and connection options) in [Response to All Reviewers - Part 1/3](https://openreview.net/forum?id=hHrO6-IfskR&noteId=15C32DfYrDA) and [Part 2/3](https://openreview.net/forum?id=hHrO6-IfskR&noteId=KwPTU0xzOrp). We will add these results to the revised version and will cite related works.
>
> > My second concern/comment is perhaps a bit unfair to this work: It is a bit baffling to me that a stateful RL algorithm like Reinforce has been widely applied (and continues to be used in the broader AutoML community) to a problem setting where one doesn't need to do long horizon credit assignment. NAS is inherently stateless, one immediately knows the result of their action (sampled architecture or in HPO the sampled hyperparameter). So 1-step RL like contextual bandits or Bayesian optimization will be just as efficient and probably more.
>
> The model we use for RL-based NAS is stateless, or to be more accurate, only has one state. The RL controller selects among multiple actions: hidden layer size choices. Policy gradient algorithms are more commonly seen to operate on a Markov-decision process (MDP) that has multiple states, in which the total reward is current state-action reward plus discounted rewards in future state-action pairs. However, this does not preclude the applicability of policy gradient to an MDP with a single state in our setting, in which the reward is indeed immediate. In fact, Williams' original 1992 paper on the REINFORCE algorithm touted the method's applicability "in both immediate-reinforcement tasks and certain limited forms of delayed-reinforcement tasks" (https://link.springer.com/content/pdf/10.1007/BF00992696.pdf, abstract).
>
> > (Following the above) Can such rejection constraints be added to BOHB, HyperBand, BANANAS, NASBO etc?
>
> We have both rejection and probability correction in our method, and have demonstrated that probability correction is necessary both in theory (Appendix H Proofs) and experiments (Appendix G Ablation studies).
>
> > If so how will they be different from those in https://arxiv.org/abs/2105.01015?
>
> The random sampling baseline we use in Figure 2 and A3 has similar rejection rules, and note that we are only sampling models close to the resource target, meaning that our baseline is stronger than simply rejecting over-constraint architectures. There’s also a difference in the underlying method: we use weight sharing. The multi-objective methods without weight sharing should be more expensive than the techniques we compare to, since they require full or partial training for many candidate architectures.
>
> **(to be continued in Part 2/2)**

---

> > ### Comment · Reviewer_7VoP · 2022-08-07
> > **Fair point about Reinforce!**
> >
> > I agree with the point about Reinforce and that it doesn't preclude the applicability to 1-step problems.
> >
> > Also acknowledging that I have carefully gone through the author's detailed responses. Thanks!!

---

> > > ### Author Response · Authors · 2022-08-07
> > > **Thanks, and regarding other questions**
> > >
> > > Thank you for taking the time to go through our response. Please let us know if we have adequately addressed your other questions/concerns as well.

---

> > > > ### Comment · Reviewer_7VoP · 2022-08-07
> > > > **Raising score!**
> > > >
> > > > I believe the paper has more merits than flaws and I like the experiments the authors have done so far. Please incorporate this response in the paper.

---

> > > > > ### Author Response · Authors · 2022-08-08
> > > > > **Thanks for the recognition**
> > > > >
> > > > > Thank you for reviewing and providing constructive feedbacks. We will for sure incorporate the response in the revised version.

---

> ### Author Response · Authors · 2022-08-02
> **Response to Reviewer 7VoP - Part 2/2**
>
> **(following Part 1/2)**
>
> > But is that necessary or one could train the supernet using …  and then doing search via any search technique?
>
> Thanks for the pointers to more refined SuperNet training techniques.
> - First, pursuing a better SuperNet is orthogonal to our goal of providing a better RL controller. We show in both the toy example (Figure 1) and on real dataset (Line 255-276) that the RL controller is to blame for suboptimal performance.
> - Nevertheless, the SuperNet training techniques we use in experiments are powerful enough for the search algorithms. To predict network performance that is used for BO and ES in Appendix E, we train a SuperNet as the following: enable all weights in the first 25% iterations, then in each iteration uniformly-at-random sample a size candidate for each layer and only do forward and backward passes on the corresponding weights. We use layer normalization in training. In this way, our SuperNet training method shares similarity with the shrinking mechanism in Once-For-All and the variance reduction techniques in https://openreview.net/forum?id=Esd7tGH3Spl. Also, we see in experiments in Appendix E that BO and ES in our current implementation get stuck at local optima in over 2/3 trials even if the SuperNet weights are representative enough to inform the existence of better architectures. Indeed, a better SuperNet + better BO and ES techniques may work better. These are directions orthogonal to our work, but are great directions for future work.
>
> > So, a comparison (on tabular task) seems orthogonal to the aim of the paper which is to show a modification of specifically Reinforce to take into account resource constraints better than others who modify the reward function?
>
> You’re right that the comparison with BO and ES seems orthogonal. We show comparison results in both of the two categories:
> - Comparison with other REINFORCE methods that modify the reward function, including TuNAS and MNasNet: This shows the proposed rejection-based RL reward finds better architectures in the feasible set.
> - Comparison with train-then-search BO and ES: This shows our RL-based one-shot weight-sharing NAS works better than train-then-search with BO or ES on our particular task.

---

### Author Response · Authors · 2022-08-02
**Response to All Reviewers - Part 1/3**

**Due to space limitations, we are splitting this section into three comments to include result tables. Sorry for the inconvenience and thank you for accommodating.**

We thank all reviewers for their careful reading and thoughtful reviews. The comments are valuable for us to improve the quality of the paper. Here, we would like to address some common comments by the reviewers. We will respond to other concerns separately under each review.

### [7VoP and w3th] Performance on the image domain

First, we would like to point out that the initial scope we intended for this paper was resource-constrained NAS on tabular datasets, as suggested by our title. This is an important and already complicated problem in research and applications, and in itself should be of great interest to the machine learning community (Line 20-25). Nevertheless, we agree with the insight of Reviewer 7VoP that the proposed method is not specific to tabular datasets. We also agree with Reviewer w3th that it would be informative to see the performance on image-based domains, and compare the rejection-based reward and the Abs Reward on the size search space of NATS-Bench [1]. This is because the co-adaptation problem we observed in Figure 1 and Section 4.1 occurs more naturally on channel sizes rather than operation and connection options.

The NATS-Bench size search space (https://arxiv.org/pdf/2009.00437.pdf, Figure 1 Top) has architectures with a predefined skeleton and 8 candidate sizes {8, 16, 24, 32, 40, 48, 56, 64} for each of its 5 layers. In this search space with 8^5=32768 candidate architectures, we use the actual #FLOPs as resource metric, validation accuracy on CIFAR-100 as RL quality reward, and test error (1 - test accuracy) on CIFAR-100 as final performance metric (the use of error instead of accuracy is consistent with other results in the paper). We use 75M #FLOPs as the resource limit, so that there are 13546 (41.3%) feasible architectures in the search space (https://arxiv.org/pdf/2009.00437.pdf, Figure 3 (e)). This experiment setting is the same as in the toy example (Figure 1), where we regard the network weights as given and only compare the RL controllers. We do grid search over NAS hyperparameters: {0.01, 0.05, 0.1, 0.5} for the RL learning rate for both the rejection-based reward and the Abs Reward, and {10, 5, 2, 1, 0.5, 0.2, 0.1} for $|\beta|$ in the Abs Reward. We show the #FLOPs and test error statistics (mean $\pm$ std) of architectures found by the rejection-based RL reward and the Abs Reward across 500 NAS repetitions for each experiment setting.

Detailed results of #FLOPs and test errors of architectures found by either of the RL rewards at different NAS hyperparameters are listed in the table below.

**#FLOPs (M):**

| RL learning rate | 0.01 | 0.05 | 0.1 | 0.5 |
|---|---:|---:|---:|---:|
| rejection-based reward, $N=200$ | 74.7 ± 0.3 (median 74.7) | 74.7 ± 0.2 (median 74.7) | 74.6 ± 0.3 (median 74.7) | 70.7 ± 4.5 (median 72.4) |
| Abs Reward, $\beta=-10$ | 77.0 ± 12.7 (median 76.4) | 75.1 ± 4.5 (median 74.8) | 75.2 ± 1.7 (median 75.1) | 75.7 ± 3.8 (median 75.2) |
| Abs Reward, $\beta=-5$ | 78.1 ± 13.1 (median 77.0) | 75.6 ± 4.3 (median 75.5) | 75.4 ± 1.7 (median 75.2) | 76.0 ± 4.4 (median 75.3) |
| Abs Reward, $\beta=-2$ | 80.6 ± 13.8 (median 78.5) | 76.3 ± 4.2 (median 76.1) | 75.7 ± 1.8 (median 75.5) | 75.9 ± 5.4 (median 75.4) |
| Abs Reward, $\beta=-1$ | 83.0 ± 13.3 (median 81.9) | 77.4 ± 4.4 (median 77.1) | 76.0 ± 2.1 (median 75.8) | 76.7 ± 5.3 (median 75.6) |
| Abs Reward, $\beta=-0.5$ | 87.1 ± 12.7 (median 86.7) | 78.8 ± 4.5 (median 78.7) | 77.0 ± 2.6 (median 76.6) | 76.9 ± 6.0 (median 76.1) |
| Abs Reward, $\beta=-0.2$ | 95.5 ± 12.8 (median 95.3) | 80.6 ± 5.2 (median 80.1) | 78.5 ± 3.6 (median 77.9) | 78.5 ± 9.2 (median 76.8) |
| Abs Reward, $\beta=-0.1$ | 101.8 ± 13.3 (median 103.0) | 84.2 ± 7.6 (median 83.0) | 81.1 ± 5.8 (median 80.5) | 80.1 ± 10.6 (median 78.6) |


**test errors:**

| RL learning rate | 0.01 | 0.05 | 0.1 | 0.5 |
|---|---:|---:|---:|---:|
| rejection-based reward, $N=200$ | 0.468 ± 0.050 | 0.435 ± 0.023 | 0.425 ± 0.014 | 0.420 ± 0.008 |
| Abs Reward, $\beta=-10$ | 0.483 ± 0.056 | 0.468 ± 0.034 | 0.473 ± 0.046 | 0.490 ± 0.074 |
| Abs Reward, $\beta=-5$ | 0.474 ± 0.049 | 0.461 ± 0.029 | 0.463 ± 0.039 | 0.481 ± 0.061 |
| Abs Reward, $\beta=-2$ | 0.462 ± 0.039 | 0.450 ± 0.022 | 0.455 ± 0.028 | 0.468 ± 0.046 |
| Abs Reward, $\beta=-1$ | 0.449 ± 0.027 | 0.442 ± 0.016 | 0.445 ± 0.020 | 0.456 ± 0.035 |
| Abs Reward, $\beta=-0.5$ | 0.436 ± 0.019 | 0.434 ± 0.014 | 0.435 ± 0.015 | 0.444 ± 0.023 |
| Abs Reward, $\beta=-0.2$ | 0.422 ± 0.013 | 0.424 ± 0.011 | 0.426 ± 0.012 | 0.433 ± 0.016 |
| Abs Reward, $\beta=-0.1$ | 0.414 ± 0.009 | 0.416 ± 0.008 | 0.418 ± 0.009 | 0.426 ± 0.014 |

**(to be continued; see Part 2/3 for the analysis on the above tables)**

---

### Author Response · Authors · 2022-08-02
**Response to All Reviewers - Part 2/3**

**(following Part 1/3)**

We can see from the #FLOPs and test error tables in Part 1/3 that:
- RL with the rejection-based reward finds architectures with confidently smaller test errors than with the Abs Reward when the #FLOPs are similar. This occurs when the Abs Reward has larger absolute values for $\beta$.
- RL with the rejection-based reward finds architectures with confidently smaller #FLOPs than with the Abs Reward when the test errors are similar. This occurs when the Abs Reward has smaller absolute values for $\beta$. Note we have a 75M #FLOPs constraint. Most architectures found by the Abs Reward exceed this constraint by >5% when they perform better than or similar to those found by the rejection-based reward.

These observations are similar to what we saw on the toy example (Figure 1 in the main paper), showing our method works across different domains.

Taking a closer look at the architectures that are Pareto-optimal and those found by each RL reward, we can see that:
- The Pareto-optimal architectures often have bottleneck structures like their counterparts in tabular tasks; examples include (in NATS-Bench format, and same below) 8:32:16:40:48, 8:56:40:56:64, and 16:56:64:64:56 that have a bottleneck in their first channel.
- Bottleneck structures occur less often in the architectures found by RL with the Abs Reward. Examples include 24:24:64:40:32, 24:40:56:40:48, and 32:48:40:32:32.
- The architectures found by RL with the rejection-based reward (our method) have more similar appearances as the Pareto-optimal architectures; examples include 8:56:64:64:64, 8:40:56:64:56, and 16:64:48:64:64.

This means the co-adaptation problem (described in Line 43 of the main paper) also occurs in RL-based NAS with resource-aware rewards in the image domain.

We want to re-emphasize that our method does not explicitly bias towards bottleneck structures; rather, it automatically learns whether a bottleneck structure is needed in an optimal architecture.

We will add the above results to the revised version.

[1] Xuanyi Dong, Lu Liu, Katarzyna Musial, Bogdan Gabrys. NATS-Bench: Benchmarking NAS Algorithms for Architecture Topology and Size. TPAMI 2021.


### [7VoP and guLP] Comparison with the RL reward that is only nonzero when resource usage is within limit (MONAS, Guerrero-Viu et al. (2105.01015), etc.).

Thanks for the pointers. Our RL reward can be understood as hard constraint + probability reweighting: we indeed sample architectures in the feasible space V as in these previous works (e.g., MONAS Equation 4), but our main novelty and the takeaway for readers is that we reweight the sampling probability $\mathbb{P}(y)$ with $\mathbb{P}(V)$, see Page 6 Equation 1. We show in Appendix G Ablation studies “Whether to differentiate through $\widehat{\mathbb{P}}(V)$” that the results are inferior if there is no probability reweighting. We will cite these works in the revised version.

**(to be continued in Part 3/3)**

---

### Author Response · Authors · 2022-08-02
**Response to All Reviewers - Part 3/3**

**(following Part 2/3)**

### [guLP and w3th] Tabulated performance of different RL rewards on datasets.

Please see the table below for a comparison among RL with the rejection-based reward, the Abs Reward, two MNasNet rewards and the reward in RENA (Resource-Efficient Neural Architect) Equation 3. Experiment settings are the same as in Appendix C and D. Bold results below indicate architectures that are on par with the best. We will add this table to the revised version.

|  | Criteo | Volkert | Aloi | Connect-4 | Higgs |
|---|---|---|---|---|---|
| reference architecture | **32-144-24 (41,153 parameters, 0.4454 ± 0.0003)** | **48-160-32-144 (27,882 parameters, 0.3244 ± 0.0040)** | 160-64-512-64 (162,056 parameters, 0.0470 ± 0.0004) | **32-22-4-20 (3,701 parameters, 0.2691 ± 0.0021)** | **16-144-16 (5,265 parameters, 0.2794 ± 0.0013)** |
| RL with rejection-based reward | **32-144-24 (2 times) (41,153 parameters, 0.4454 ± 0.0003) 32-112-32 (40,241 parameters, 0.4456 ± 0.0003)** | **64-128-48-16 (27,050 parameters, 0.3237 ± 0.0040) 48-160-32-144 (27,882 parameters, 0.3244 ± 0.0040) 80-48-112-32 (27,802 parameters, 0.3274 ± 0.0037)** | **176-144-144-80 (2 times) (161,672 parameters, 0.0458 ± 0.0007) 192-112-176-80 (161,432 parameters, 0.0461 ± 0.0012)** | 20-32-14-48 (3,701 parameters, 0.2827 ± 0.0062) 20-30-24-22 (3,693 parameters, 0.2795 ± 0.0049) 28-10-12-56 (3,701 parameters, 0.2819 ± 0.0052) | 72-24-48 (2 times) (5,161 parameters, 0.2911 ± 0.0031) 80-32-8 (5,265 parameters, 0.2893 ± 0.0022) 64-16-128 (5,265 parameters, 0.2870 ± 0.0033) |
| RL with Abs Reward | 32-64-96 (2 times) (41,345 parameters, 0.4461 ± 0.0003) 32-80-64 (40,785 parameters, 0.4459 ± 0.0002) | 96-64-32-48 (27,738 parameters, 0.3302 ± 0.0042) 96-48-32-96 (27,738 parameters, 0.3305 ± 0.0047) 96-80-16-48 (27,738 parameters, 0.3302 ± 0.0050) | 144-112-144-96 (3 times) (162,008 parameters, 0.0473 ± 0.0004) 128-96-96-112 (4 times) (162,072 parameters, 0.0488 ± 0.0012) 112-112-96-112 (161,816 parameters, 0.0502 ± 0.0007) | **30-22-12-12 (3,703 parameters, 0.2702 ± 0.0027)** 26-20-18-26 (2 times) (3,703 parameters, 0.2807 ± 0.0029) 26-18-20-26 (3,703 parameters, 0.2765 ± 0.0046) | 104-16-24 (2 times) (5,233 parameters, 0.2896 ± 0.0036) 80-16-88 (2 times) (5,281 parameters, 0.2887 ± 0.0021) 88-16-64 (2 times)  (5,217 parameters, 0.2874 ± 0.0027) |
| MNasNet (reward type 1: $Q(x) (T(x) / T_0)^\beta$) | 32-64-16 (36,065 parameters, 0.4464 ± 0.0003) 32-64-8 (35,537 parameters, 0.4466 ± 0.0002) 32-24-64 (35,353 parameters, 0.4466 ± 0.0003) | only found one feasible architecture: 32-32-224-24 (19,890 parameters, 0.3392 ± 0.0043) | 160-128-112-96 (163,544 parameters, 0.0469 ± 0.0009) 128-96-96-112 (162,072 parameters, 0.0488 ± 0.0012) 128-176-128-80 (153,192 parameters, 0.0473 ± 0.0007) | 24-18-16-44 (3,677 parameters, 0.2788 ± 0.0046) 28-12-32-14 (3,651 parameters, 0.2781 ± 0.0052) 22-32-16-22 (3,577 parameters, 0.2818 ± 0.0032) | 72-24-32 (4,745 parameters, 0.2898 ± 0.0019) 64-32-16 (4,545 parameters, 0.2888 ± 0.0018) 72-24-24 (2 times) (4,537 parameters, 0.2903 ± 0.0036) |
| MNasNet (reward type 2: $Q(x) \max \\{1, (T(x) / T_0)^\beta \\}$) | 24-128-16 (29,953 parameters, 0.4468 ± 0.0005) 16-176-48 (27,985 parameters, 0.4478 ± 0.0006) 24-16-160 (27,953 parameters, 0.4479 ± 0.0003) | only found one feasible architecture: 80-24-24-24 (17,874 parameters, 0.3521 ± 0.0013) | 112-144-240-64 (145,944 parameters, 0.0480 ± 0.0005) 160-112-128-64 (126,392 parameters, 0.0497 ± 0.0019) 256-64-224-32 (104,232 parameters, 0.0548 ± 0.0015) | 20-12-44-24 (3,679 parameters, 0.2871 ± 0.0069) 28-16-10-52 (3,745 parameters, 0.2776 ± 0.0061) 14-40-28-20 (3,581 parameters, 0.2886 ± 0.0047) | 16-8-16 (2 times) (777 parameters, 0.2929 ± 0.0020) 8-8-24 (553 parameters, 0.3003 ± 0.0031) |
| RENA | 32-16-48 (34,289 parameters, 0.4472 ± 0.0002) 24-48-160 (33,873 parameters, 0.4468 ± 0.0001) 8-64-96 (15,137 parameters, 0.4523 ± 0.0004) | only found one feasible architecture: 32-48-24-512 (26,482 parameters, 0.3389 ± 0.0039) | didn’t find any (close to) feasible architecture | 24-26-20-16 (3,617 parameters, 0.2806 ± 0.0068) 30-18-10-32 (3,749 parameters, 0.2752 ± 0.0075) 26-26-18-8 (3,577 parameters, 0.2769 ± 0.0031) | only found one feasible architecture: 80-8-152 (4,569 parameters, 0.2882 ± 0.0016) |

**(end of Response to All Reviewers)**

---

### Meta-Review · Area_Chair_6SRK · 2022-08-27

**Recommendation:** Accept
**Confidence:** Certain

**Metareview:**

The reviewers unanimously recommend accept.  Many important clarification points were discussed in the author rebuttal.  Please make sure to incorporate those changes into the final camera ready.

**Award:**

No

---

### Decision · Program_Chairs · 2022-09-14

Accept